# FORECASTING NEEDLES IN A TIME SERIES HAYSTACK

## ABSTRACT

Shocks and sudden spikes are common characteristics of real-world time series data. For example, demand surges or electricity outages often occur in time series data, manifesting as spikes ("Needles") added to the regular time series ("Haystack"). Despite their importance, it is surprising to find their absence in the benchmarking protocol at the frontier of time series research—Time Series Foundation Models (TSFM). To address this gap, we present the Needle-in-a-Time-Series-Haystack (NITH) Benchmark, which includes both synthetic and real-world spiky time series data from diverse domains like traffic, energy, and biomedical systems. For synthetic data, we develop a flexible framework using Poisson-based modeling to generate spiky time series, allowing us to evaluate forecast models under various conditions. To accurately assess model performance, we introduce a new metric based on Dynamic Time Warping, specifically designed for spiky data. We evaluate the zero-shot forecasting capabilities of six popular TSFMs over 64 million observations, identifying their limitations related to architecture, tokenization, and loss functions. Furthermore, we demonstrate that the incorporation of the proposed NITH dataset, due to its diversity compared to the common pre-training corpus of TSFMs, results in improved performance.

## 1 INTRODUCTION

Time series forecasting, where future events are predicted given the past, is a critical task with applications in many industries, such as climate science, traffic engineering, and advertising. Forecasting spiky time series is particularly important, because rare events, such as solar flares, traffic accidents and holiday sales, are numerically represented as narrow spikes occurring sparsely across a long time series. Due to their rarity and short duration, we refer to these spikes as 'needles' and the base time series without spikes as the 'haystack'. The ability to accurately forecast needles in a times-series haystack gives practitioners the foresight to plan around rare and significant events.

Recently, foundation models for time series forecasting (TSFMS) (Woo et al., 2024; Goswami et al., 2024; Das et al., 2023; Ansari et al., 2024; Liu et al., 2024b) have garnered significant attention. Unlike traditional time series forecasting models (Rangapuram et al., 2018; Wang et al., 2019; Salinas et al., 2020; Rasul et al., 2023; Lim et al., 2021), foundation models do not require training on downstream datasets to perform effective inference. Instead, they first pretrain on a large corpus of general time series, then perform zero-shot forecasting on any downstream univariate time series. The development of foundation models has democratized time series forecasting from only individuals with large training datasets to the general public. Given the success of TSFMS on standard benchmarks and the significance of spiky time series forecasting, our work studies the zero-shot effectiveness of TSFMS on spiky time series specifically.

Although forecasting spikes is known to be challenging (Ansari et al., 2024), existing zero-shot forecasting model only measure model performance in aggregate and not specific classes of time series. Hence, it is unclear whether such models can reliably forecast spikes and when it cannot. In contrast, our study systematically identifies some problem areas current TSFMS struggle with, offering suggestions for developing the next generation of TSFMS. To the best of our knowledge, we are the first to study TSFMS' performance on spiky time series data.

Our work introduces two times series benchmarks, a **Real**-world **N**eedle **i**n a **T**ime-series **H**aystack dataset (NITH-REAL), and a **Synth**etic **N**eedle **i**n a **T**ime-series **H**aystack dataset (NITH-SYNTH). For the real-world datasets, we collect multiple spiky time series datasets, across various do-

mains (Dau et al., 2018; Liu et al., 2024b; Lavin & Ahmad, 2015) to evaluate whether existing foundation models can immediately be used in practice for spiky time series forecasting. For the synthetic datasets, we control the underlying generating process behind spiky time series to study TSFM sensitivity to specific needle properties. Generally, spikes either occur following some underlying pattern, such as periodicity (Martin & Mailhes, 2010), or are ad-hoc, occurring due to random noise (Wu & Keogh, 2021). Because it is difficult to distinguish between complex patterns and noise, our synthetic dataset explicitly models the 2 factors independently to measure what properties of spiky time series TSFM struggle most with, outside of noise.

Because spikes occur over short durations and are often acompanied by small amounts of random lag, we propose a lag-aware metric, based on dynamic time warping (Müller, 2007), to accurately benchmark TSFM performance on spiky datasets. After benchmarking existing TSFM on NITH-REAL and NITH-SYNTH, our work additionally provides a synthetic training dataset that slightly improves zero-shot performance on real-world datasets. Finally, our work benchmarks different model and dataset design choices, finding that pretraining dataset plays the largest role in spiky time series forecasting. In summary, our contributions are:

- We create a real-world (NITH-REAL) and a synthetic (NITH-SYNTH) benchmark, which measures the ability to forecast spiky time series of different TSFMs. Our curated real world and synthetic data has 22M and 42M observations respectively, amassing 64M. Besides, our synthetic can be used for benchmark and continual pretraining.

- To generate a controlled synthetic benchmark, we propose a novel and flexible mathematical framework. Continual pretraining with our proposed synthetic datasets can significantly improve real-world performance, demonstrating that our NITH-SYNTH resembles real-world data.

- We systematically benchmark six popular TSFMs on our NITH dataset and examine when TSFMs fail to forecast spiky time series, finding all tested foundation models fail at forecasting narrow needles.

- To further dissect this problem, we evaluate different dataset, model, and loss function choices finding that dataset design plays the largest role in spiky time series forecasting performance. We also conclude context length is not the current bottleneck for spiky time series forecasting.

## 2 RELATED WORK

### 2.1 TIME SERIES FOUNDATION MODELS

Recently, TSFMs have been proposed for *zero-shot* forecasting on numerical time series, outperforming traditional time series forecasting models such as ETS, ARIMA (Hyndman et al., 2008), and Theta (Assimakopoulos & Nikolopoulos, 2000) under univariate settings. Early TSFMs used prompting to query language models for forecasting tasks (Jin et al., 2023; Gruver et al., 2024; Chang et al., 2023). Instead of prompting, PatchTST (Nie et al., 2022) uses patching (Alexey, 2020) to directly tokenize and finetune on the time series. Later works experiment with different tokenization schemes, such as LagLlama's (Rasul et al., 2023) lag-based covariate tokenizer and Chronos' (Ansari et al., 2024) quantile-based sample tokenizer. By pretraining on larger datasets, these models exhibit effective zero-shot abilities. Other works utilize updated model architectures, such as TimeFM's (Das et al., 2023) decoder-only architecture and Moment's (Goswami et al., 2024) encoder-decoder architecture. Certain works, like Moment (Goswami et al., 2024), require finetuning to adapt masked token reconstruction pretraining to the forecasting task. Moirai (Woo et al., 2024) open-sourced a large real-world pretraining dataset, LOTSA, which were then used by later works such as Timer (Liu et al., 2024b) to perform tasks beyond univariate forecasting. TinyTMS (Ekambaram et al., 2024) allows efficient forecasting though smaller transformers. TimeGPT (Garza & Mergenthaler-Canseco, 2023) provides commercial API access to a closed-source model for zero-shot forecasting. In this work, we test 6 recent open-source time series forecasting models (Rasul et al., 2023; Ansari et al., 2024; Das et al., 2023; Woo et al., 2024; Liu et al., 2024b; Ekambaram et al., 2024). While newer TSFMs tackle tasks beyond univariate forecasting or improve inference efficiency, our work questions whether TSFM can really perform effective zero-shot forecasting on a particularly important class of univariate time series, spiky time series.

## 2.2 Time Series Benchmarks and Data

Many TSFMs originate their data from the Monash repository (Godahewa et al., 2021), the M3 M4 benchmarks (Godahewa et al., 2021), and ETT (Zhou et al., 2021). Moirai (Woo et al., 2024) and Timer (Liu et al., 2024b) collected and cleaned these datasets into the LOTSA and UTSD pretraining datasets respectively. Although M3 and M4 were originally time series benchmarks themselves, they now belong in the pretraining data of many TSFM (Woo et al., 2024). Chronos (Ansari et al., 2024) uses synthetic data in its training, and TimesFM (Das et al., 2023) uses close-source datasets. Under the zero-shot setting, Timer offers several benchmarks that do not overlap with LOTSA and UTSD, including a larger set of time series from the UCR anomaly benchmark (Wu & Keogh, 2021). In addition to UCR, Numenta also offers several time series in its anomaly detection benchmark (Lavin & Ahmad, 2015). Beyond zero-shot univariate time series forecasting, several recent works study non-general time series forecasting performance (Bauer et al., 2021; Godahewa et al., 2021; Wang et al., 2024c; Qiu et al., 2024), and TSFMs' zero-shot multimodal (Liu et al., 2024a) and infilling (Liu et al., 2024b) capabilities. This work is the first to study TSFM's zero-shot forecasting capabilites on spiky time series.

## 3 Needle-In-a-Time-series-Haystack Benchmark

### 3.1 Time Series Forecasting

We study the forecasting problem on univariate spiky time series. Specifically, given a time series with a context of $C$ steps, $u_x = [u_0, u_1, ..., u_{C-1}]$, our goal is to forecast the next $F$ steps, $u_y = [u_C, u_{C+1}, ..., u_{C+F-1}]$. We benchmark time series forecasting models, $\mathcal{M}$, which predict the next $F$ steps by tokenizing the context, passing it through a transformer, then detokenizing the time series. We denote inference by $\tilde{u}_y = \mathcal{M}(u_x)$, where $\tilde{u}_y$ is the predicted forecast.

#### 3.1.1 Spiky Time Series

Spiky time series are time series with repeated spikes. When these spikes follow some underlying pattern, it is possible for forecasting models to forecast them[1]. Compared to standard time series, forecasting spiky time series induces 3 new challenges: (1) spikes occur infrequently within the context, thus the model has limited information, (2) spikes occur only for a short duration, thus the model must memorize past occurrences, and (3) the average duration between spikes is subject to some noise, thus the model must be robust. We call these challenges the "difficulty dimensions."

Specifically, given $N_{spikes}$ spikes that occur at indices, $\mathcal{J} = [\hat{\tau}_0, \hat{\tau}_1, ..., \hat{\tau}_{N_{spikes}}]$, we quantify each difficulty dimension by (1) the average duration between spikes, $\mu_T = \sum_{i=0}^{N_{spikes}} \frac{\hat{\tau}_i}{N_{spikes}}$, (2) the width of spikes, $\omega^2$, and (3) standard deviation in duration between spikes, $\sigma_T = \sqrt{\sum_{i=0}^{N_{spikes}} \frac{(\hat{\tau}_i - \mu_T)^2}{N_{spikes}}}$, respectively.

Because forecasting spiky time series requires the model to learn from the past spikes, which both occur infrequently and over short durations, our proposed benchmark tests the memory ability of forecasting model. Thus, we call our problem "needle in a haystack", where "needles" are spikes and the "haystack" is the base time series without the spikes. We note that the terms "needle" and spike are interchangeable in the rest of this work. We call our benchmarks "*N*eedle-*I*n-a-*T*ime-series-*H*aystack" (NITH), consisting of real, NITH-REAL, and synthetic, NITH-SYNTH, time series.

### 3.2 NITH: Real-World Spiky Time Series

We construct a real-world spiky time series benchmark, NITH-REAL, by filtering existing benchmarks, which contain both spiky and non-spiky time series. Because peaks are spike-shaped patterns

---

[1] We highlight our work is separate from outlier detection, where outlier occurrences do not follow an underlying pattern, instead being treated as noise, which cannot be forecast.

[2] Because adding a spike to a base time series is non-injective, it is impossible to recover the width of the spike, $\omega$, from an already spiky time series. Thus, we provide separate definitions for width when dealing with an already spiky time series, $\omega'$, in Section 3.2 and the generative process of a spiky time series, $\omega$, in Section 3.3.

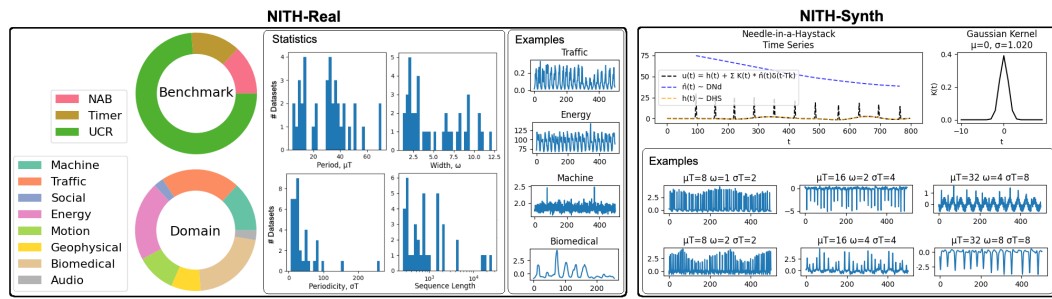

Figure 1: NITH Benchmark. On the left, we show statistics about NITH-REAL with some time series visualizations. Full dataset statistics are in Appendix J. On the right, we show how to generate NITH-SYNTH with some time series visualizations.

that occur throughout the time series, we identify needle indices, $I$, by running a peak detection algorithm, and define the width, $\omega'$, as the peak detection width (Virtanen et al., 2020; Sci, 2018). Because any forecasting model predicts future time steps based on an appropriate context, we filter only datasets where peaks occur at least 8 times on average across a context window of $C = 512$ over each time series. In addition, we ensure that at least one spike occurs in both the forecast window of $F = 128$. We apply our filtering algorithm on the NAB benchmark (Lavin & Ahmad, 2015), UCR Anomaly benchmark (Wu & Keogh, 2021), and Timer's test set (Liu et al., 2024b). In total, NITH-REAL consists of 38 datasets, $> 7$K time series, and $> 22$M observations from a wide variety of domains and cover a wide range of the difficulty dimensions (Figure 1). We provide the full benchmark statistics in Appendix J.

### 3.3 NITH: SYNTHETIC SPIKY TIME SERIES

We motivate our synthetic dataset, NITH-SYNTH, by the limited size and scope of real-world datasets and by observing that real-world spikes may either be the product of some underlying pattern or noise. To train and evaluate on more NITH-style data, we provide a mathematical framework for generating diverse spiky time series. To disentangle the contribution of noise from underlying patterns, our framework follows the most simple pattern, periodicity, where a spike occurs every $\mu_T$ time steps "in-expectation" with standard deviation of $\sigma_T$. Note, although this pattern is seemingly simple, no TSFMs could accurately forecast needles across all settings.

Specifically, we define spiky time series in NITH-SYNTH benchmark by enforcing the following properties: (1) **"outlier constraint"**: the needle must have larger amplitude than the haystack, (2) **"rarity constraint"**: the needle must be near-nonzero only for a small fraction of the time series, and (3) **"periodic-in-expectation"**: a spike occurs every $\mu_T$ time steps "in-expectation" with standard deviation of $\sigma_T$. The summarized generation algorithm can be found at Alg 1.

We mathematically model a "periodic-in-expectation" spiky signal as a Markov chain transitioning between a haystack signal, $h(t) : \mathbb{R} \to \mathbb{R}$, and a needle signal, $n(t) : \mathbb{R} \to \mathbb{R}$. In this work, we consider additive needles, $n(t) = h(t) + \hat{n}(t)$, where both the residual, $\hat{n}(t) : \mathbb{R} \to \mathbb{R}$, and haystack, $h(t)$, are independently sampled from Gaussian processes, $\mathcal{GP}_{Nd}(\mu(t), k(t,t'))$ and $\mathcal{GP}_{HS}(\mu(t), k(t,t'))$ respectively. Specifically, one sample from the needle is observed every $\tau_i \sim \mathcal{D}_T$ time steps of the haystack. Under this formulation, the needle indices are $\mathcal{J} = [\hat{\tau}_i]_{i=0}^{N_{samples}} = [\sum_{j=0}^{i} \tau_j]_{i=0}^{N_{samples}}$, and the resulting signal would be: $h(t) + \sum_{i=0}^{N_{samples}-1} \hat{n}(t)\delta(t - \hat{\tau}_i)$, where $\delta(t - \hat{\tau}_i)$ is a Dirac-Delta function at time $\hat{\tau}_i$.

Because needles do not occur instantaneously, we model transients by convolving the needle observations with a with a Kernel function, $\mathcal{K}(t)$, and define the width, $\omega$, as the bounds containing 95% of the area: $\int_{-\frac{\omega}{2}}^{\frac{\omega}{2}} \mathcal{K}(t)dt \geq 0.95 \int_{-\infty}^{\infty} \mathcal{K}(t)dt$. In practice, we use the Gaussian kernel, $\mathcal{K}_{\sigma_\mathcal{K}}(t) = \frac{1}{\sqrt{2\pi\sigma_\mathcal{K}^2}} \exp\left(-\frac{x^2}{2\sigma_\mathcal{K}^2}\right)$. In summary, we (1) sample a haystack signal from a Gaussian Process, (2) sample a schedule on where to add needles, (3) sample a needle signal from a separate Gaussian Process, (4) convolve the needle signal with a kernel according to the schedule, and (5) add the needle and haystack signals, as formalized in Equation 1:

$$u(t) = h(t) + \Sigma_{i=0}^{N_{samples}-1} \mathcal{K}(t) * \hat{n}(t)\delta(t - \Sigma_{j=0}^{i}\tau_j) \tag{1}$$

**Outlier Constraint.** We quantify the outlier constraint by considering needles with larger RMS amplitude than the haystack: $RMS_n \gg RMS_h$, where $RMS_x = \sqrt{\frac{1}{L}\int_0^L x^2(t)\,dt}$ and $[0, L)$ is the duration of interest. We enforce the outlier constraint by scaling by $\alpha \sim U(3, 5)$, DC offset by $\beta \sim U(1, 5)\alpha$, and flipped by $\theta \sim \{-1, +1\}$, the resulting signals sampled from the needle Gaussian process $\mathcal{GP}_{Nd}(\mu(t), k(t, t'))$ to obtain $\hat{n}(t)$. We directly sample the haystack signal: $h(t) \sim \mathcal{GP}_{HS}(\mu(t), k(t, t'))$.

**Rarity Constraint.** We quantify the outlier constraint by considering narrow needles: $\mu_T \gg \omega$. We enforce the rarity constraint by setting the width to be smaller than the period $\omega \leq 0.25\mu_T$. This can be accomplished by tuning the standard deviation of the kernel using the Gaussian inverse Q-function (Karagiannidis & Lioumpas, 2007): $\omega = 2Q^{-1}(\frac{1-0.95}{2})\sigma_{\mathcal{K}}$.

**On Randomness: Periodic-in-Expectation.** The Poisson distribution, $Pois(\lambda)$ naturally models periodic signals in expectation with period, $\mu_T = \lambda$. However, the said distribution's standard deviation is a strictly a function of its mean, $\sqrt{\lambda}$. We can flexibly set the standard deviation by sampling the distribution at different sampling periods, $\Delta$, where $\mathcal{K}_i = \mathcal{K}(i\Delta)$. Specifically, the standard deviation is dispersed, $\sigma_T$, when the sampling rate is $\Delta = \frac{\mu_T}{\sigma_T^2}$ and distribution parameter is set to $\lambda = \Delta\mu_T$. We provide the proof in Appendix B.

---

**Algorithm 1** Code to Generate NITH Datasets

**Input:** $GP_{Nd}, GP_{HS}, \mu_T = N_{occ}, \omega, \sigma_T, \alpha, \beta, \theta, \Delta, C, F$
**Output:** $[u_k]_{k=0}^{C+F-1}$
$h(t) \sim GP_{HS}$
$\hat{n}_{raw}(t) \sim GP_{Nd}$
$\hat{n}(t) \leftarrow \theta(\alpha\hat{n}_{raw}(t) + \beta)$
$\sigma_{\mathcal{K}} \leftarrow \frac{\omega}{2Q^{-1}(\frac{1-0.95}{2})}$
$T_0, T_1, ..., T_{N_{samples}-1} \sim \mathcal{N}(\mu_T, 0.1\mu_T)$
$[\hat{T}_{k'}]_{k'=0}^{N_{samples}} \leftarrow [\Delta \cdot \lfloor T_{k'}/\Delta \rfloor]_{k'=0}^{N_{samples}}$
$u(t) \leftarrow h(t) + \Sigma_{k=0}^{N_{samples}-1}\sigma_{\mathcal{K}}\mathcal{K}_{\sigma_{\mathcal{K}}}(t) * \hat{n}(t)\delta(t - \Sigma_{k'=0}^{k}\hat{T}_{k'})$
$[u_k]_{k=0}^{C+F-1} \leftarrow [u(k\Delta)]_{k=0}^{C+F-1}$

---

By choosing an appropriate sampling rate, $\Delta$, we disentangle the expected duration, $\mu_T$, from the randomness, $\sigma_T$. We call this the scaled-Poisson distribution, $SPois(\mu_T, \sigma_T)$. Purely periodic spikes are generated when $\sigma_T = 0$, and purely random spikes are generated when $\sigma_T = \infty$. We set the "haystack" distribution, $\mathcal{D}_T$ as a the scaled Poisson distribution $\mathcal{D}_T = SPois(\mu_T, \sigma_T)$.

### 3.3.1 CONSIDERED SETTINGS

To cover a diverse range of spiky time series, we generate 100 time series of length 5000 for each combination of periods of $\mu_T = [8, 16, 32, 64]$, widths of $\omega = [\frac{\mu_T}{4}, \frac{\mu_T}{8}, \frac{\mu_T}{16}, \frac{\mu_T}{32}, \frac{\mu_T}{64}]$, and noise of $\sigma_T = [0, 1, 2, 4, 8, 16]$. Skipping duplicate settings, NITH-SYNTH consists of 840 datasets, with 84K time series, and 42M observations.

### 3.4 EVALUATION METRICS

### 3.4.1 PROBLEMS WITH EXISTING METRICS

Evaluation metrics are an important component for benchmarking time-series data. Typically, time series forecasting models are evaluated on aggregated errors such as Mean Absolute Error (MAE) or Mean Squared Error (MSE). Because MAE and MSE treat each sample independently and the haystack accounts for a greater proportion of samples than the needle, aggregated errors both overemphasizes the haystack and cannot measure signal lag. Another common metric is Dynamic Time Warping (DTW) (Müller, 2007), which computes the optimal alignment between 2 time series, $DTW(\tilde{u}, u) = d_{DTW}(\tilde{u}_{N_{needle}}, u_{N_{needle}})$ where $d_{DTW}(\tilde{u}_i, u_j) = C[i, j] + min(d_{DTW}(u_{i-1}, s_j), d_{DTW}(u_i, s_{j-1}))$, according to some cost function $C[i, j] = ||\tilde{u}_i - u_j||_2^2$. Because DTW is lag-invariant and the haystack accounts for a greater proportion of samples than the needle, DTW both overemphasizes the haystack and cannot effectively measure lag between true

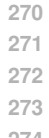
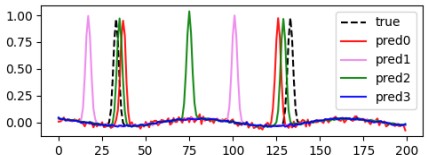
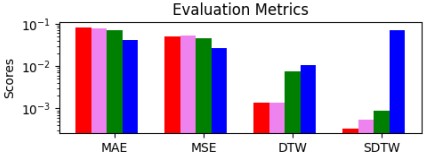

(a) Case Study Signals. "true" signal is black and "pred" signals are colored.

(b) Evaluation metrics. Bar color represents error between "true" and corresponding "pred" signal.

Figure 2: Evaluation Metric Case Study: We measure the error between "true" and "pred" signals. Visually, "pred0" most closely matches "true", followed by "pred1" and "pred2", followed by "pred3." SDTW is the only metric that captures this ordering.

and predicted needles. In terms of classification, precision and recall (Tatbul, 2018; Hwang et al., 2019) adequately measures lag and and classification performance. However, this metric cannot quantify the regression differences in spike magnitudes, which our benchmark aims to study.

### 3.4.2 SCALED DYNAMIC TIME WARP

To effectively evaluate spiky time series, we propose a scaled Dynamic Time Warping (SDTW) metric, which combines the recall and false positive rate with dynamic time warping.

**Scaling DTW for Needle (Recall):** To compute DTW over the true needles, we first define the probability that each sample, $u_i$, in the time domain corresponds with a needle, $p_{needle}(i|\mathcal{J})$, where $\mathcal{J} = [\hat{\tau}_0, \hat{\tau}_1, ..., \hat{\tau}_{N_{spikes}}]$ represents ground truth needle locations. To focus on "just needles", we accumulate cost only across the true needle by multiplying the cost matrix by the probability of the predicted signal belongs to a needle: $C_{SDTW}^{(needle)}[i,j] = p_{needle}(j|\mathcal{J})C_{DTW}^{(needle)}[i,j]$. We quantify the probability that each sample, $u_i$, in the time domain corresponds with a needle through a mixture of Gaussians distribution: $p_{needle}(i|\mathcal{J}) = p_{mix}(k)p_{comp}^{(needle)}(i|k,\mathcal{J})$. Specifically, we define whether an index corresponds with the $n$-th needle as $p_{comp}(i|k,\mathcal{J}) \propto \mathcal{N}_{PDF}(J_k, \sigma_{SDTW})(i)$, where $\mathcal{N}_{PDF}(\cdot, \cdot)$ defines the probabilistic distribution function of the Gaussian distribution, scaled such that the needle indices are guaranteed to be the needle $p_{needle}(J_k|\mathcal{J}) = 1$. We define the mixture distribution as uniform, $p_{mix}(k) = \frac{1}{N_{needle}}$, and scale the standard deviation to the mean period $\sigma_{SDTW} = \frac{\mu_T}{16}$.

**Scaling DTW for Haystack (False Positive Rate):** To compute DTW over the true haystack, we consider the inverse probability as the needle $p_{haystack}(i|\mathcal{J}) = 1 - p_{needle}(i|\mathcal{J})$. To focus on "just haystack", we accumulate cost across the predicted haystack by multiplying the cost matrix by the probability of the predicted signal belongs to a needle: $C_{SDTW}^{(haystack)}[i,j] = p_{haystack}(i|\mathcal{J})C_{DTW}^{(needle)}[i,j]$. We multiply over the predicted signal instead of the true signal here, such that any needles predicted in what should be the haystack will accumulate cost.

**SDTW:** In summary, we compute the SDTW between predicted time series, $\tilde{u}$, and true time series, $u$, as follows: $SDTW(\tilde{u}, u) = d_{SDTW}^{(needle)}(\tilde{u}_{N_T}, u_{N_T}) + d_{SDTW}^{(haystack)}(\tilde{u}_{N_T}, u_{N_T})$, where $d_{SDTW}^{(setting)}(\tilde{u}_i, u_j) = C_{SDTW}^{(setting)}[i,j] + min(d_{SDTW}(u_{i-1}, s_j), d_{SDTW}(u_i, s_{j-1}))$. In practice, we perform z-score normalization using the context's statistics prior to computing SDTW and empirically observe models fail to forecast needles when SDTW is greater than 2.0.

We provide a case-study showing the benefits of SDTW over MAE, MSE, and DTW. As shown in Figure 2, "pred0" should perform better than the other 3 models. "pred2" ignores the needles entirely, but fits the haystack perfectly. "pred2" predicts the needle with long lag-times before the true needle, but fits the haystack slightly better than "pred0". "pred3" predicts extra needles. Because MAE and MSE treats each sample evenly, it pays more attention to the haystack, artificially ranking "pred2" higher. Because DTW is invariant to lag, artificially ranking "pred2" higher. SDTW correctly identifies "pred0" as the most similar signal to "true," by both allowing some lag in the signal and weighing the needle and haystack equally. **We provide a more detailed case study around SDTW's inner workings in Appendix D.**

# 4 RESULTS

In this section, we study the performance of existing time series foundation models on spiky time series, then ablate both model and dataset design choices to determine performance bottlenecks. We benchmark 6 open-source time series transformer foundation models: TIMER (Liu et al., 2024b), MOIRAI (Woo et al., 2024), TINYTS (Ekambaram et al., 2024), CHRONOS (Ansari et al., 2024), LAGLLAMA (Rasul et al., 2023), and TIMESFM (Rasul et al., 2023). We chose these foundation models as they cover a wide variety of design choices [3], which we list in Appendix C. Our study focuses on zero-shot performance, as the primary objective of pretraining large foundation models is to enable inference without additional training.

## 4.1 MAIN RESULTS

Our results on NITH-REAL (Figure 3) shows all TSFM fail on the majority of datasets. Because no TSFM can predict the needle with an with standard error bounds below 2.0, all models fail on the majority of datasets. Large models that have been pretrained on more data tend to perform better, with TIMESFM, which was trained on $> 50B$ samples, achieving the best performance among unmodified TSFM. Intuitively, a larger pretraining corpus cover more diverse signals, such as spiky time series. Unlike TSFM, large language models are trained on datasets at the trillion scale. Our results indicate a greater need for high-quality open-source time series training data.

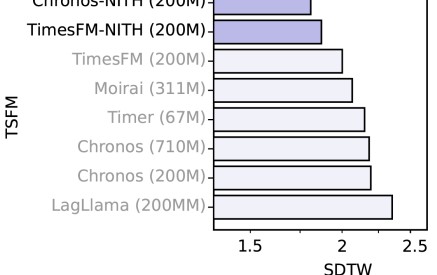

Figure 3: TSFM zero-shot performance on NITH-REAL benchmark. Models with the "-NITH" suffix were continuously pretrained on our synthetic dataset. Among vanilla TSFM, TIMESFM achieve the best performance out-of-the-box. For readability reasons, we show TINYTS, which performs poorly, in Appendix E.

Given that pretraining data seems to be a bottleneck for TSFM performance on NITH, we perform **continuous pretraining** on the 2 most effective TSFM from Section 4.2: CHRONOS and TIMESFM. Specifically, we use our synthetic dataset generation process, outlined in Algorithm 1 to generate additional pretraining data for these TSFM under a different random seed and $\sigma_T = 0.0$. We then perform continual training on this dataset, to enhance CHRONOS and TIMESFM. We call the resulting models CHRONOS-NITH and TIMESFM-NITH. We use the same training scripts as provided by the official CHRONOS (Ansari et al., 2024) and TIMESFM (Das et al., 2023) codebases.

We quantify the effect of noise on the pretraining process by both pretraining CHRONOS on a synthetic corpus generated using $\sigma_T = 0.0$ and one generated using $\sigma_T \sim 2^{U(0, log_2(\mu_T))}$, which covers the range of values used in Synthetic NITH. Compared to the latter, the former is less noisy but has a distribution mismatch with the synthetic NITH dataset. Pretraining on noisy datasets (Figure 8) greatly deteriorates model performance, even when there is no distribution mismatch. As stated in Section 3.4, aggregated loss functions such as MSE and Cross Entropy treats each sample independently, hence any lag induced by noise is not properly modeled by the loss. This is particularly concerning as real-world data is particularly noisy (Figure 1). Thus, more effort on pretraining dataset selection is required to effectively train time series foundation models to forecast needles.

Continual pretrained TSFM on synthetically-generated NITH data outperforms the best TSFM on real-world NITH data, under the zero-shot setting (Figure 3). These results suggest that in lieu of curated real-world pretraining data, synthetically-generated NITH data can boost performance of existing foundation models. We note that parameters for synthetic NITH can be flexibly tuned for specific applications, but leave such study to future work.

---

[3]Although some modeling decisions have been ablated on generic time series, there lacks a systematic study over these model choices, specifically with relation to forecasting spiky time series.

## 4.2 ON NEEDLE PROPERTIES

To further dissect this problem, we investigate what type of data existing TSFM struffle with us. Evaluating TSFM on synthetic NITH-SYNTH with across different difficulty dimensions (Figure 4), we reveal the width of the needle has a larger effect than the period between needle occurances. it is evident by the fact that we observe a significant improvement across columns with different $\omega$ but the same $\mu_T$. As expected, needles that occur more randomly, with larger $\sigma_T$, are also harder to forecast. However, certain TSFM are more resilient to such noise. Specifically, CHRONOS' performance decays slightly less than TIMESFM and MOIRAI in presence of noise. In contrast, TIMESFM performance decays slightly less than CHRONOS and MOIRAI at forecasting narrow needles. The exact reason for these phenomenon are hard to determine, as each TSFM differs across multiple design decisions.

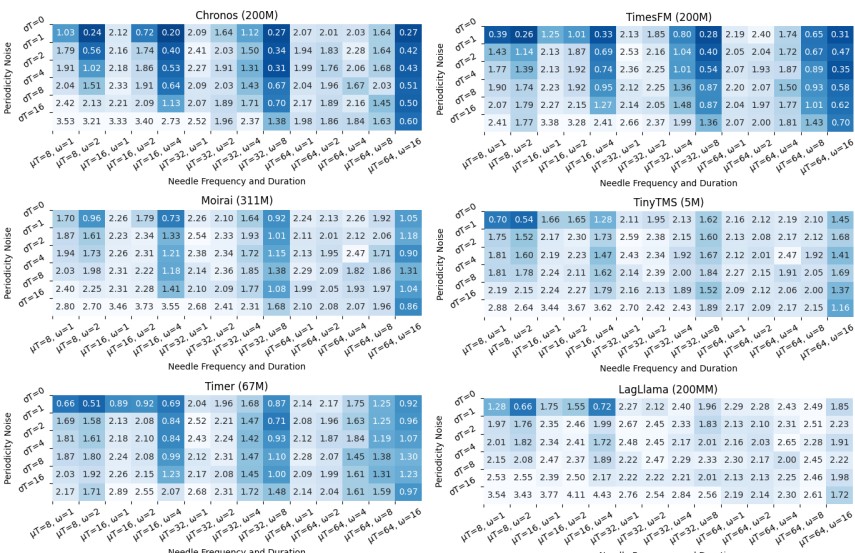

Figure 4: TSFM's zero-shot performances on NITH-SYNTH. The x-axis denotes the period between spikes, $\mu_T$, and the width of spikes, $\omega$. The y-axis denotes the noise, $\sigma_T$. Numbers represent average SDTW value across a particular setting. TIMESFM and CHRONOS perform the best. We show results on the largest CHRONOS (710M) model, which performs similarly to CHRONOS (200M), in Appendix F.

To quantify the current capabilities of TSFM, we observe all models fail when the standard deviation in noise is more than half the period, $\sigma_T > \frac{\mu_T}{2}$, or the needle occurs over less than an eigth of the period, $\omega < \frac{\mu_T}{8}$. This results shows that spiky datasets is a failure case for existing TSFM, justifying the need for NITH benchmark. We visualize these failure cases (Figure 6), observing TSFM completely ignores narrow needles. We hope our findings spur further research in TSFM to overcome such barriers.

## 4.3 ON CONTEXT LENGTH

Some TSFM's are trained with longer context lengths to better forecast long time series. In theory, TSFM's with longer context could observe more needles, improving performance. To test the effect of this phenomenon, we increase the inference-time context lengths of TSFM pretrained with longer context from 512: MOIRAI, TIMER, and LAGLLAMA. Specifically, we try the longest context-lengths each model was pretrained on and the smallest context length LAGLLAMA was trained on, following each its recommended settings (Rasul et al., 2023).

Evaluating TSFM's pretrained on larger context length with different inference-time context lengths (Figure 5) shows long context has little effect on NITH performance. As explained in Section 4.1, TSFM's cannot remember narrow needles at all. Hence, context length is not a current bottleneck for TSFM performance on spike time series. Instead, different choices in tokenizer hyperparameters,

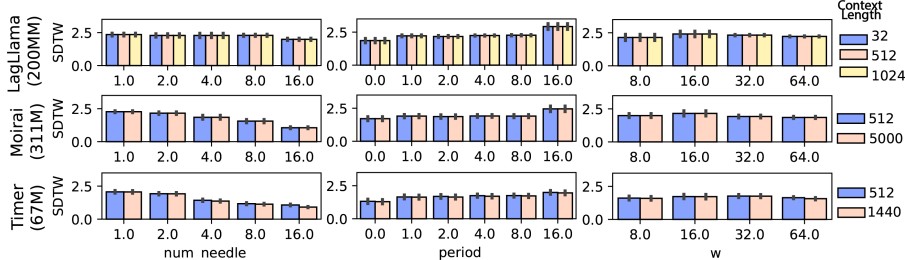

Figure 5: TSFM with different context length's aggregated performances on NITH-SYNTH, under different settings of $\mu_T$, $\sigma_T$, and $\omega$. We provide a line chart in the Appendix. Context lengths have minimal affect of performance.

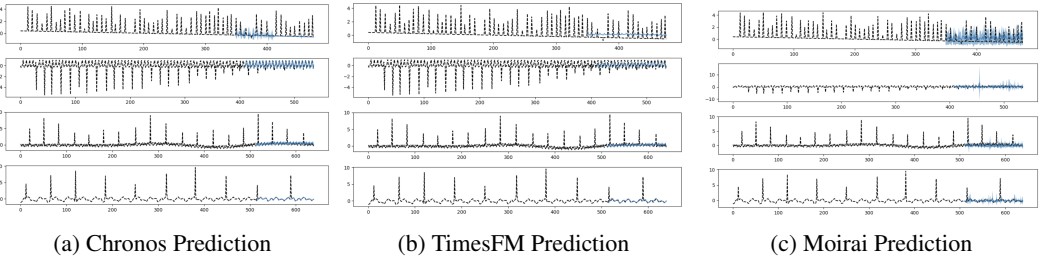

(a) Chronos Prediction          (b) TimesFM Prediction          (c) Moirai Prediction

Figure 6: Performance of different foundation models on narrow needle setting. All foundation models fail to recover the needle signal.

model architecture, loss function, and pretraining dataset between the 3 tested TSFM has a much larger impact, which we further discuss in Appendix H.

## 4.4 ON MODEL DESIGN

Finally, we provide further insight into tokenizer design decision for TSFM, by evaluating CHRONOS pretrained from scratch on synthetic NITH with $\sigma_T = 0$, as motivated in Section 4.1, with different tokenizer settings. Specifically, we benchmark TSFM performance with different patch-lengths and model architectures. As a reference, TIMESFM uses a patch length of 32. Given the positive performance of CHRONOS, we ablate smaller patch lengths of 8, 16, and 32. In this study, we consider when stride is equal to the patch length. We leave further analysis to future work. We consider encoder, encoder-decoder, and decoder only models with fixed number of layers and dimension size. Our results hold with fixed number of layers and larger dimension size as well.

**Architecture.** We begin our investigation by testing the architecture across the most popular tokenizer setup, patch-based tokenizers. By evaluating SDTW performance on the synthetic NITH benchmark (Figure 7), we find that encoder-decoder outperform both encoder-only and decoder-only architectures. We hypothesize this is because encoder-decoder models combine bidirectional attention, extracting more information from the context, with autoregressive generation, which generate tokens conditioned on past predictions. Patch length has a smaller overall impact than architecture on SDTW performance. Hence, it is worthwhile to first ablate architecture-level decisions when training new TSFM.

**Patch Length.** As shown in Figure 7, shorter patch lengths outperform longer patch lengths when forecasting needles. Because patch-based tokenizers are convolutional layers, they low-pass filter the data with a kernel-size equal to the patch-length. Because, the larger patch-length correlates with more aggressive filtering, smaller patch lengths learn narrower needles better. We believe automatically choosing the correct kernel-size for downstream problems could greatly improve efficacy of TSFM. Some existing TSFM, such as MOIRAI, accomplish this by pretraining with multiple patch-based tokenizers.

**Tokenizer.** We ablate the best patch-based tokenizer with the top-performing sample-based tokenizer, CHRONOS's tokenizer. Our work shows that sample-based tokenizers performed better (Figure 8), as sample-based tokenizers do not implicitly perform any low-pass filtering on the down-

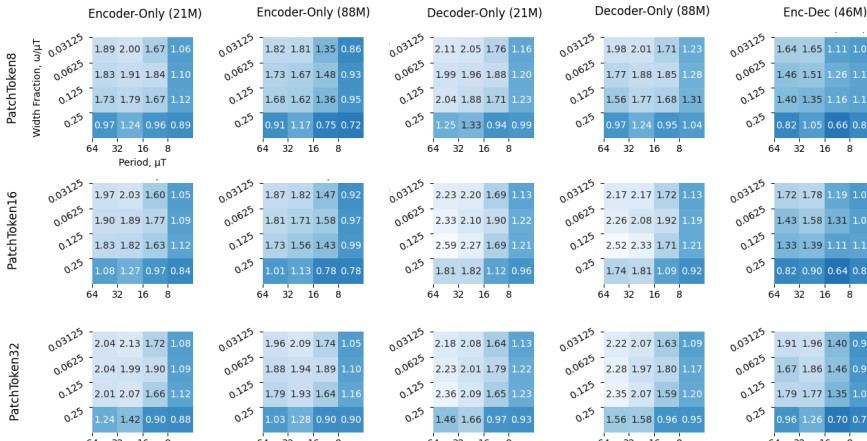

Figure 7: Testing different patch tokenizer and architectures across the $\sigma_T = 0.1\mu_T$ setting of NITH-SYNTH. Small patch lengths perform the best at forecasting spiky time series. Because train and test datasets use the same generative process (except train set uses $\sigma_T = 0.0$) with different random seed, there is no distribution shift.

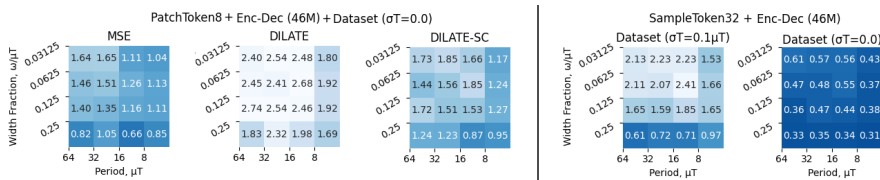

Figure 8: Testing different loss function and dataset choices across the $\sigma_T = 0.1\mu_T$ setting of NITH-SYNTH. Noiseless training data is critical for forecasting spiky time series. Because train and test datasets use the same generative process with different random seed, there is no distribution shift.

stream dataset. However, the top performing sample-based tokenizer uses quantile-binning which cannot represent needle values if they occur too infrequently across the signal (Ansari et al., 2024). Overcoming this bottleneck could scale sample-based tokenizers to narrower needles with smaller $\omega$. We believe tokenization is a major bottleneck that can unlock effective TSFM for NITH-style time series.

**Loss Function** To test whether alignment-based loss functions can improve upon aggregation-based loss, we try training the best patch-based tokenizer with DILATE, a differentiable version of DTW loss. We notice that DTW tends to produce unnatural alignments on the haystack portion of the signal leading to poor convergence, shown in Appendix D. For this reason, we also try DILATE with a Sakoe-Chiba window constraint of window size 32, which we dub DILATE-SC. Although DILATE and DILATE-SC still does not outperform MSE loss (Figure 7), we believe alignment-based losses are a promising direction for spiky time series forecasting.

## 5 CONCLUSION

In conclusion, while existing time series foundation models made remarkable progress in zero-shot univariate time series forecasting on most workloads, our analysis has identified substantial challenges these models face when forecasting spiky time series. Specifically, we have identified three core issues that existing foundation models encounter: 1) the average duration between spikes, 2) the width of the spikes, and 3) the randomness in duration between spikes. To systematically study each weakness and real-world uses, we proposed NITH-SYNTH and NITH-REAL benchmarks.

## 6 ETHICS STATEMENT

Our work does not involve human subjects. Our work does not introduce harmful data, as our benchmarks are either curated collections of existing open-source data or synthetically generated time series. Our benchmark does not release any harmful insights, methodologies, applications, conflicts of interest and sponsorship, discrimination/bias/fairness concerns, privacy and security issues, legal compliance, and research integrity issues.

## 7 REPRODUCIBILITY

We cite tested foundation models in Section 4 and provide aby additional hyperparameter settings in Appendix I. provide NITH-REAL's dataset source and statistics in Appendix J. We provide NITH-SYNTH's settings in Section 3.3.1 and psuedocode for synthetic dataset generation in Algorithm 1 . We will open source our benchmark code once the paper is accepted.

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

| | Pretrain Data | Context Size | Tokenizer | Architecture | Model Size |
|---|---|---|---|---|---|
| TIMESFM | 100B | $\leq 512$ | Patch | Dec | 200M |
| MOIRAI | 27.65B | $\leq 5000$ | Patch | Enc | 311M |
| TIMER | 28B | $\leq 1440$ | Patch | Dec | 67M |
| CHRONOS | 84B | $\leq 512$ | Sample | Enc-Dec | 710M |
| LAGLLAMA | 0.36B | $\leq 1024$ | Sample | Dec | 200M |
| TINYTS | 1B | $\leq 1536$ | Patch | Enc-Dec | 5M |

Table 1: Statistics of different time series foundation models.

Haoyi Zhou, Shanghang Zhang, Jieqi Peng, Shuai Zhang, Jianxin Li, Hui Xiong, and Wancai Zhang. Informer: Beyond efficient transformer for long sequence time-series forecasting. In *Proceedings of the AAAI conference on artificial intelligence*, volume 35, pp. 11106–11115, 2021.

## A    LIMITATIONS

Our work benchmarks open-source time series foundation models (TSFM) on forecasting both a real-world and synthetic spiky time series. We leave benchmarking closed-source TSFM, particularly TimeGPT (Garza & Mergenthaler-Canseco, 2023) to future work. Our synthetic datasets model the simplest underlying pattern: periodic in expectation. Because current TSFM cannot solve even the simple case, we leave benchmarking more complex forms of periodicity (Martin & Mailhes, 2010), to future work. Our work provides general insights and suggestions on model, dataset, and loss design. We leave extensive hyperparameter tuning to verify our general recommendations to future work. This work collects time series data from related works rather than providing our own new dataset. Hence, we depend on generous open-source real-world data to conduct our study.

## B    SCALED POISSON DISTRIBUTION PROOF.

We show that by sampling the Poisson distribution with sampling period $\Delta$, we can arbitrarily scale the mean, $\mu_T$, and standard deviation, $\sigma_T$. Specifically, we consider both the sampled Poisson, $\hat{p}(n)$, and the original Poisson distribution, p(t), below:

$$p(t) = \frac{\lambda^t e^{-\lambda}}{t!}$$

$$\hat{p}(n) = \frac{\lambda^{n\Delta} e^{-\lambda}}{(n\Delta)!}$$

We find the mean is scaled by the sampling period:

$$\mu_T = \mathbb{E}_{\hat{p}}[n] = \mathbb{E}_p[\frac{t}{\Delta}]$$

$$\mu_T = \frac{\lambda}{\Delta}$$

Next, we find the sampling rate scales the variance as a function of the mean duration.

$$\sigma_T = \sqrt{Var_{\hat{p}}[n]}$$

$$\sigma_T = \sqrt{Var_p[\frac{t}{\Delta}]}$$

$$\sigma_T = \frac{1}{\Delta}\sqrt{Var_p[t]}$$

$$\sigma_T = \frac{1}{\Delta}\sqrt{\lambda}$$

$$\sigma_T = \sqrt{\frac{\mu_T}{\Delta}}$$

Hence, we can specify any $\mu_T$ and $\sigma_T$ by setting $\lambda$ and $\Delta$. In summary, we define the scaled Poisson distribution, $SPois(\mu_T, \sigma_T)$, by its probability density function, $\hat{p}(n)$, where $\lambda = \frac{\mu_T^2}{\sigma_T^2}$ and $\Delta = \frac{\mu_T}{\sigma_T^2}$.

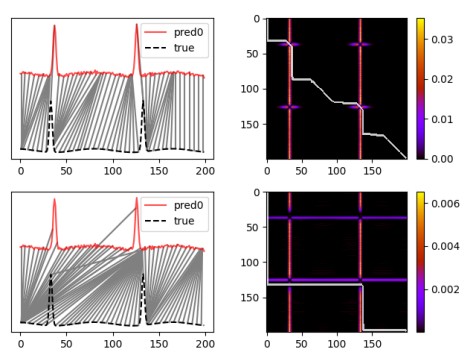

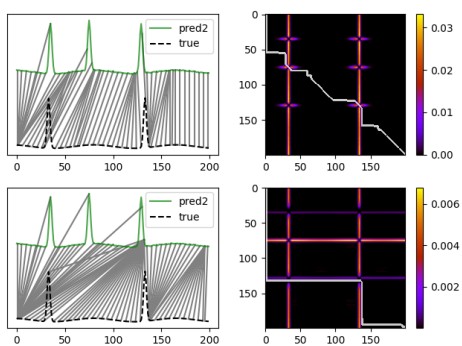

(a) Example SDTW time domain matching and cost matrices between "pred0" and "true."

(b) Example SDTW time domain matching and cost matrices between "pred2" and "true."

Figure 9: SDTW interpretability: We plot the time domain matchings and cost matrix associated with the needle, $C_{SDTW}^{(needle)}$, (top), and haystack, $C_{SDTW}^{(haystack)}$, (bottom). As expected, extra needles are heavily punished. Some lag between predicted and true needles is tolerated.

## C    TSFM DIFFERENCES

In Table 1, We list the differences between time series foundation models (TSFM). Note, no 2 models change only 1 design parameter between them, hence it is difficult to ascertain the exact reasons one TSFM performs better than another.

## D    SDTW CASE STUDY

To further analyze the SDTW metric, we visualize the DTW paths in Figure 9, which corresponds to the time series in Figure 2. The horizontal lines correspond with the predicted needles and verticle lines correspond with true needles. In the top 2 plots, predicted needles (horizontal lines) are masked out over the haystack. Hence, the aggregated SDTW metric measures whether predicted needle occurs when the true needle does (recall). In the bottom 2 plots, predicted needles (horizontal) lines are masked out over the needle indices. Hence, the aggreagted SDTW metric measures whether the predicted needle occurs over the haystack (false positive rate).

## E    NITH-REAL FULL RESULTS

We show TSFM performance, including TINYTS, on each dataset of NITH-REAL in Figure 10. Note, TINYTS performs very poor skewing the plot. We believe this is due to TINYTS being much smaller than baselines.

## F    NITH-SYNTH FULL RESULTS

For fair comparison, we showed models of roughly equal size in the main text. In Figure 11, we show the largest CHRONOS (710M) model achieves similar performance as the base CHRONOS (200M) model suggesting that performace improvements from model size may not be a bottleneck. We believe higher quality pretraining data is needed to further improve the TSFM with increasing model size.

## G    TSFM PREDICTION VISUALIZATIONS

We visualize the TSFM predictions in Figures 14 to 35.

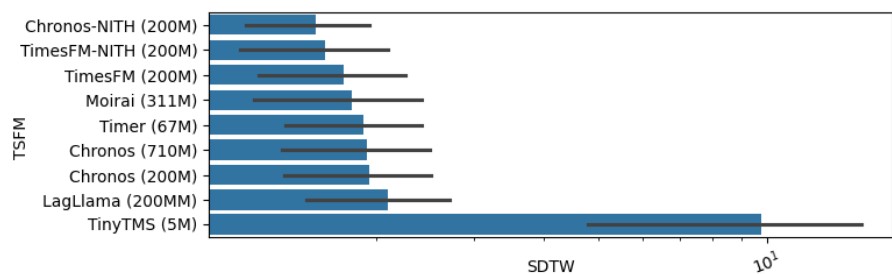

Figure 10: TSFM zero-shot performance on NITH-REAL benchmark. Models with the "-NITH" suffix were continuously pretrained on our synthetic dataset. Among vanilla TSFM, TIMESFM achieves the best performance out-of-the-box. We show TINYTS in this version. Error bars denote standard error.

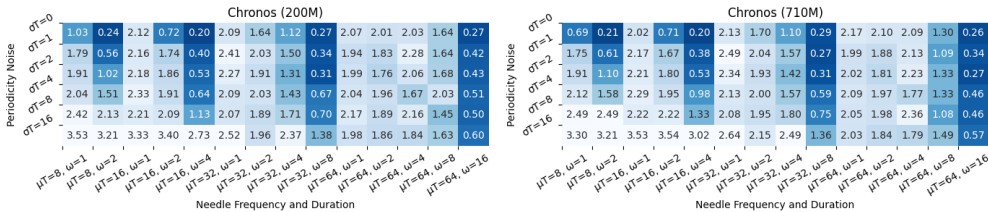

Figure 11: CHRONOS' zero-shot performances on NITH-SYNTH with different model sizes. The x-axis denotes the period between spikes, $\mu_T$, and the width of spikes, $\omega$. The y-axis denotes the noise, $\sigma_T$. Numbers represent average SDTW value across a particular setting.

## H    FULL CONTEXT LENGTH RESULTS

We provide the full context length results in Figure 12. We also aggregate into a line plot representation in Figure 13. From these plots, we observe model performance more related to model architecture than context length.

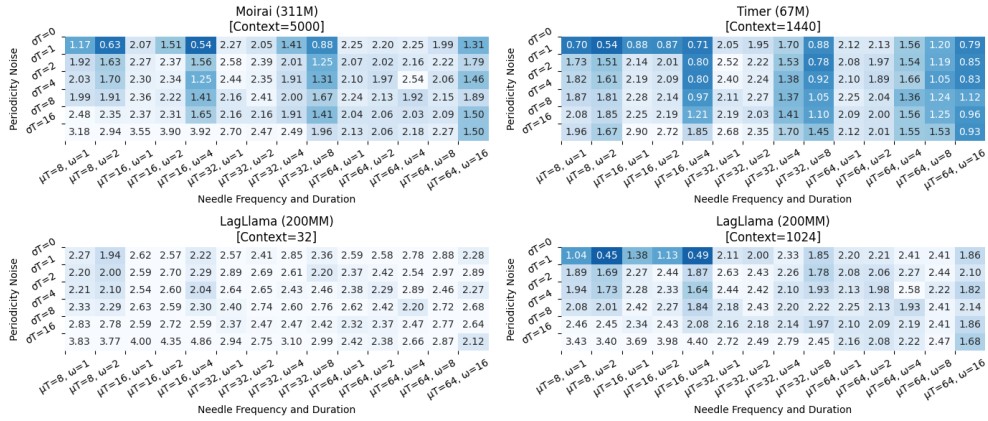

Figure 12: TSFM's zero-shot performances on NITH-SYNTH with different context lengths. The x-axis denotes the period between spikes, $\mu_T$, and the width of spikes, $\omega$. The y-axis denotes the noise, $\sigma_T$. Numbers represent average SDTW value across a particular setting.

## I    HYPERPARAMETER SETTINGS

We follow the default hyperparameter settings for TSFM models, except with a forecast size of 128 and a trajectory count of 20 for models supporting probabilistic forecasting. We found minimal em-

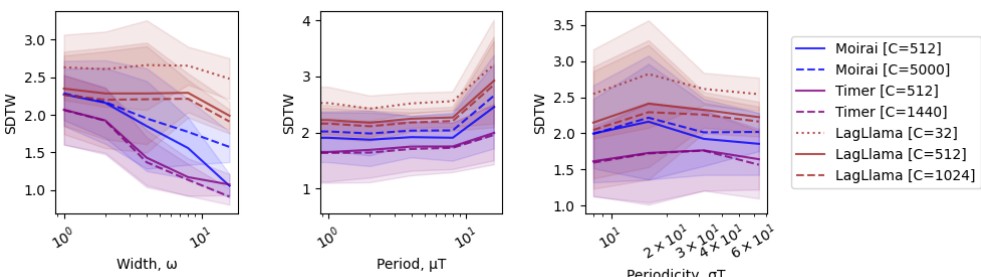

Figure 13: TSFM with different context length's aggregated performances on NITH-SYNTH, under different settings of $\mu_T$, $\sigma_T$, and $\omega$. Line chart version. Context lengths have minimal affect of performance.

pirical difference between using 20 trajectories compared to using 1 trajectory. For training -NITH models, we adopt the default pretraining setup as CHRONOS and finetuning setup as TIMESFM. We provide dataset statistics and generation configurations for $\mu_T$, $\omega$, and $\sigma_T$ in the main text.

## J DATASET STATISTICS

We provide the dataset statistics for the real-world dataset in Tables 2 and 3. We determined the domain qualitiatively by manually evaluating the source of each dataset. We provide aggregated statistics in the main text.

## K ADDITIONAL RELATED WORKS

### K.1 NEEDLE-IN-A-HAYSTACK

Needle-in-a-haystack search can be approached using various techniques, broadly categorized into classical methods and machine learning-based approaches. Classical methods include heuristic search techniques, such as brute-force search, which examines every element in the dataset, and index-based methods, which improve efficiency by pre-processing the dataset to quickly locate potential matches (Chauhan & Vig, 2015). However, these methods often struggle with high-dimensional data and complex patterns (Pang et al., 2021).

Specifically, machine learning models such as convolutional neural networks (CNNs) and recurrent neural networks (RNNs) have been applied to rare object detection in images, and RNNs have been used for anomaly detection in sequential data. Additionally, transformer-based models, which excel at handling long-range dependencies in data, have been applied to this problem, offering improved performance in tasks such as DNA sequence analysis (Beltagy et al., 2020; Zaheer et al., 2020; Kitaev et al., 2020; Gu & Dao, 2023). To characterize the language recalling ability of large language models (Tay et al., 2020; Gu et al., 2021; 2022), many needle-in-a-haystack benchmarks have been proposed (Hengle et al., 2024; Wang et al., 2024a;b; Kuratov et al., 2024; Zhao et al., 2024; Gu & Dao, 2023). In general, recalling information that occurs infrequently in the dataset is a tough but reliable challenge to measure the strengths of different foundation models. Here we propose a new needle-in-a-haystack dataset in the time series domain. In addition to that, our work investigates how to forecast needles instead of detecting needles in a retrieving manner.

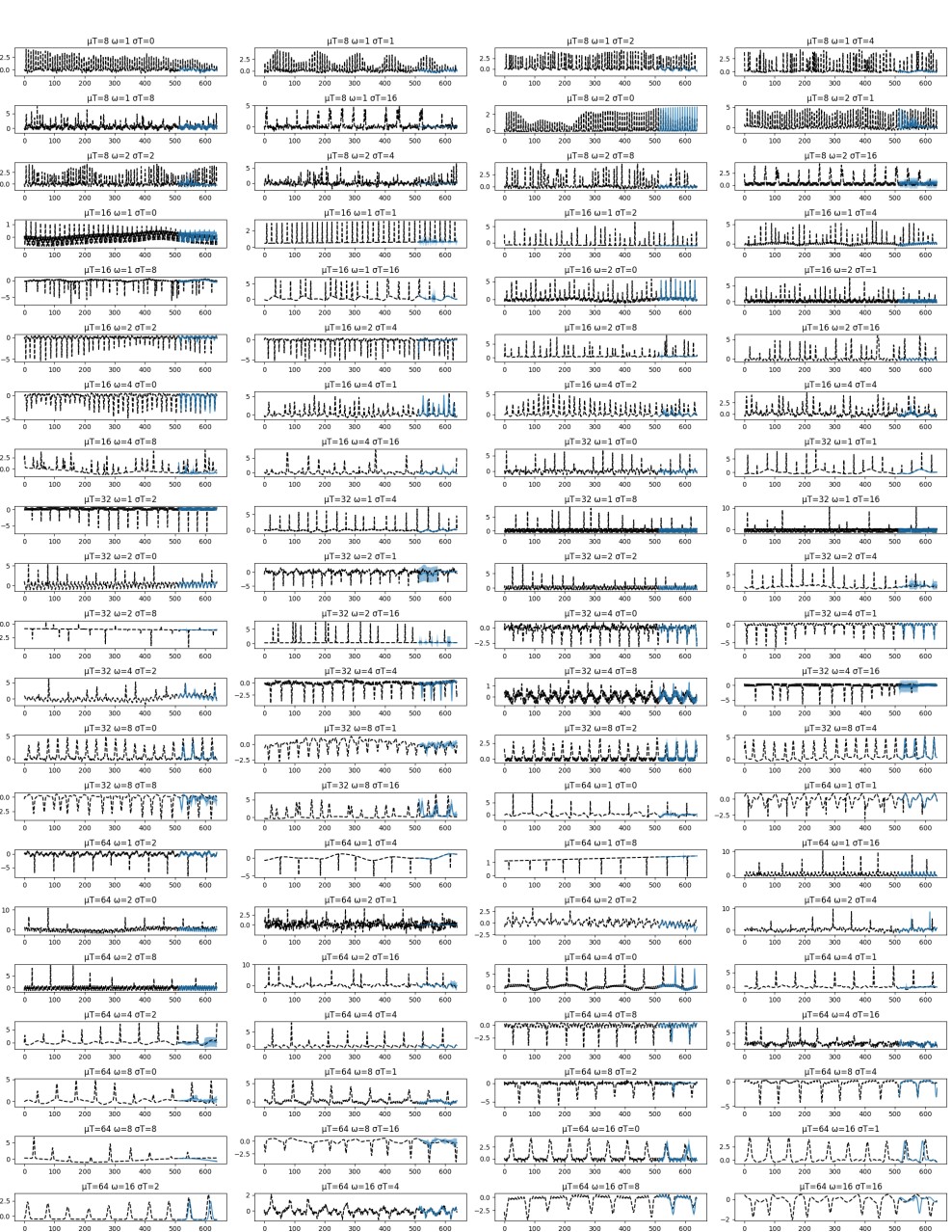

Figure 14: CHRONOS (200M) predictions on NITH-SYNTH.

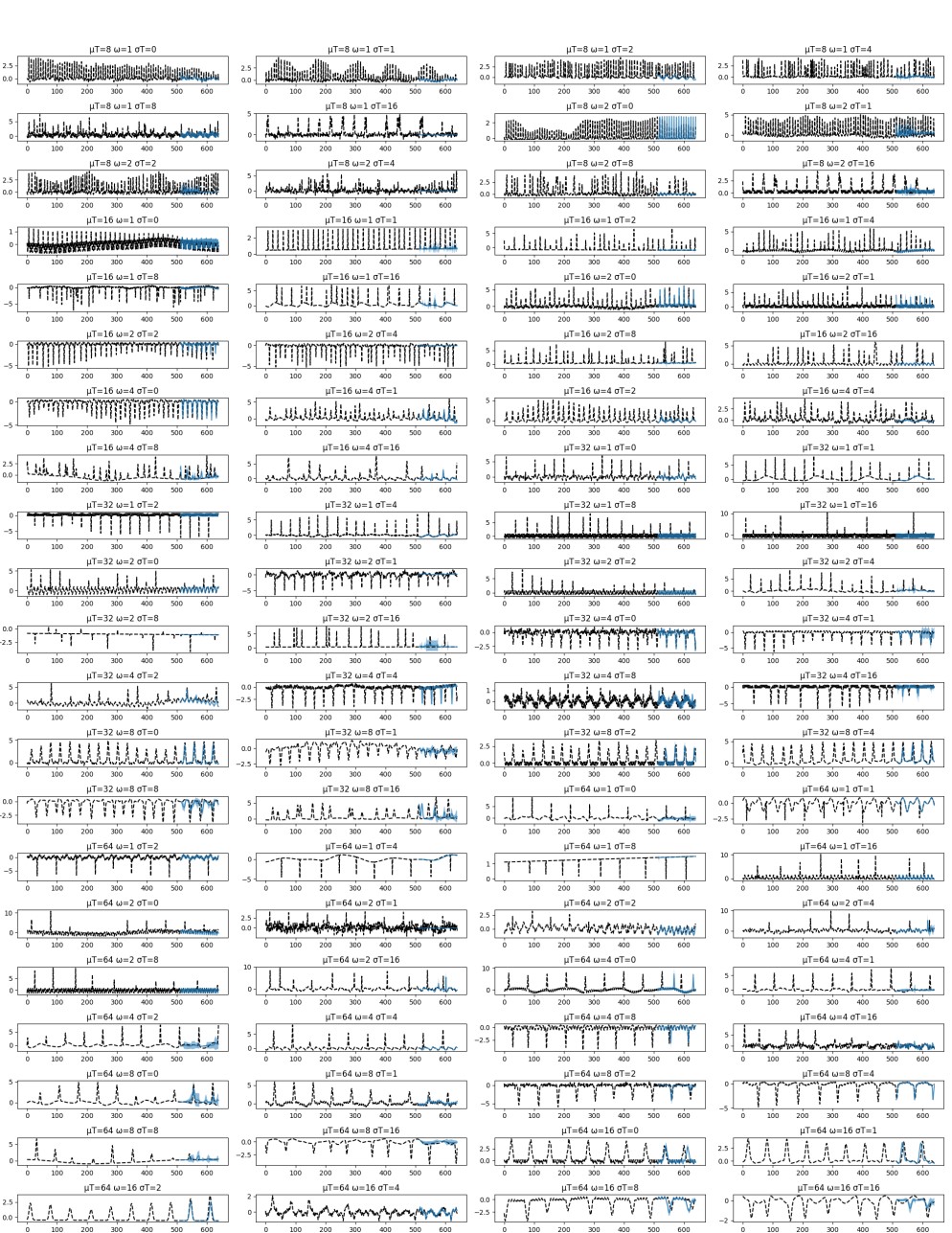

Figure 15: CHRONOS (710M) predictions on NITH-SYNTH.

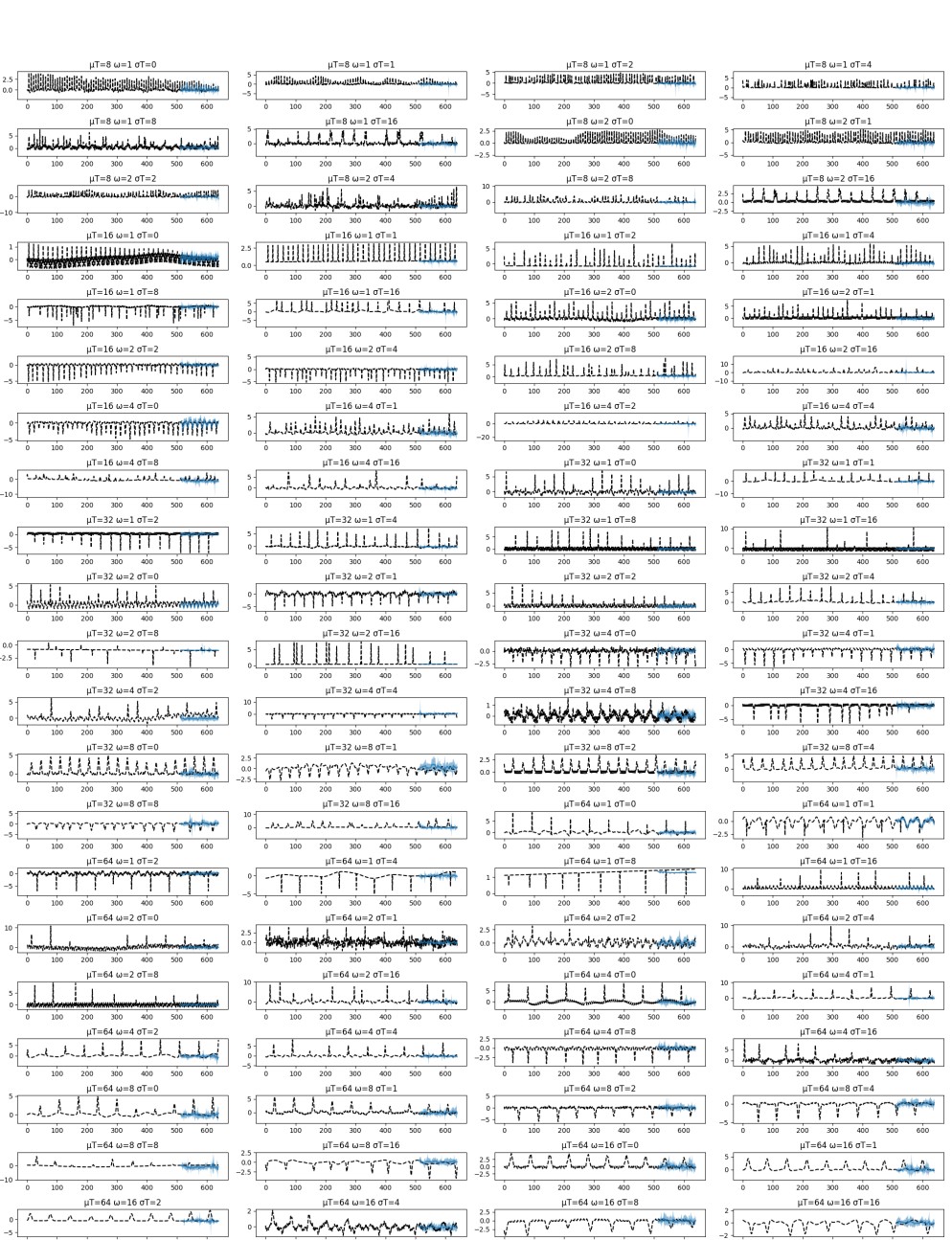

Figure 16: LAGLLAMA (200M) predictions on NITH-SYNTH.

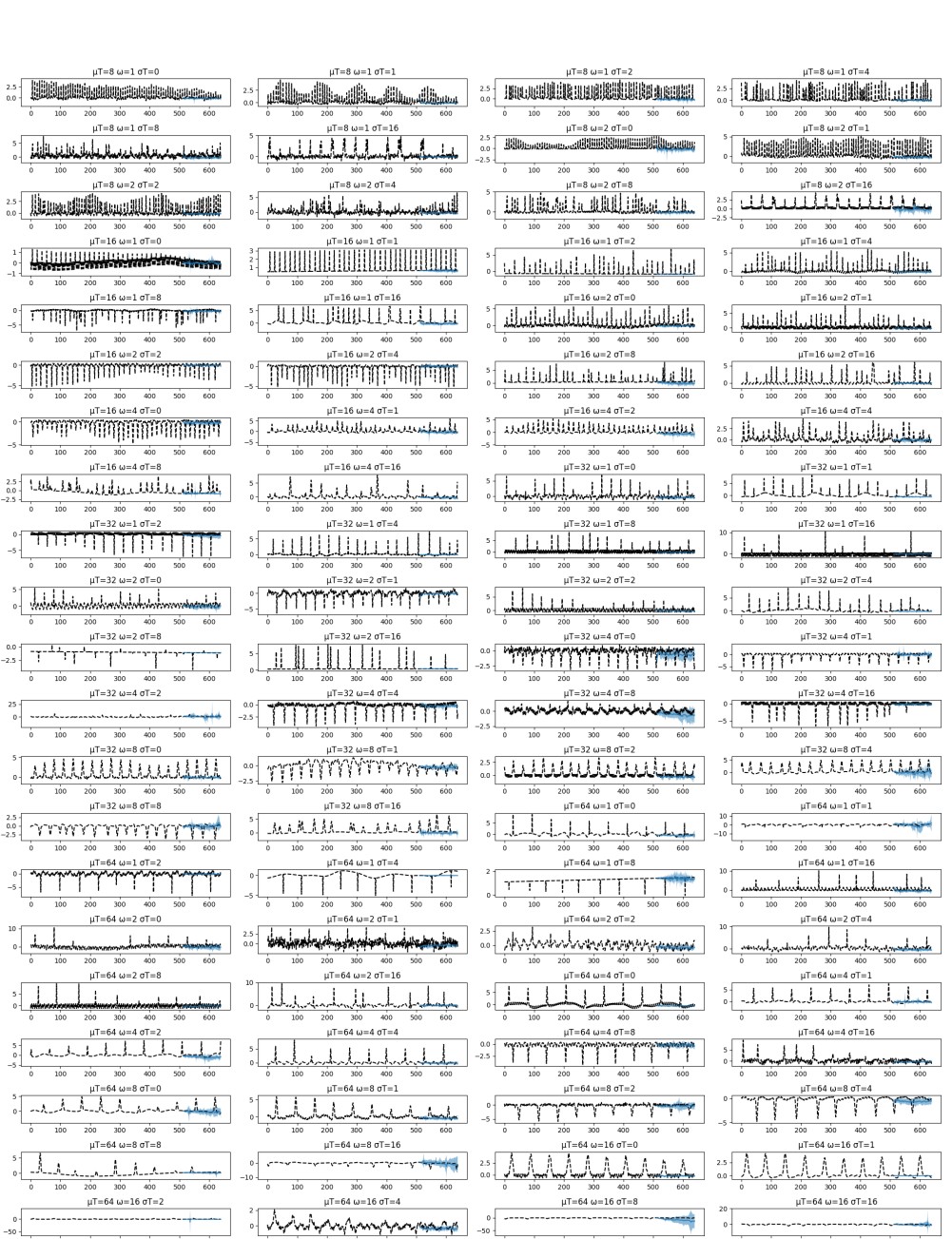

Figure 17: LAGLLAMA (200M) [Context=32] predictions on NITH-SYNTH.

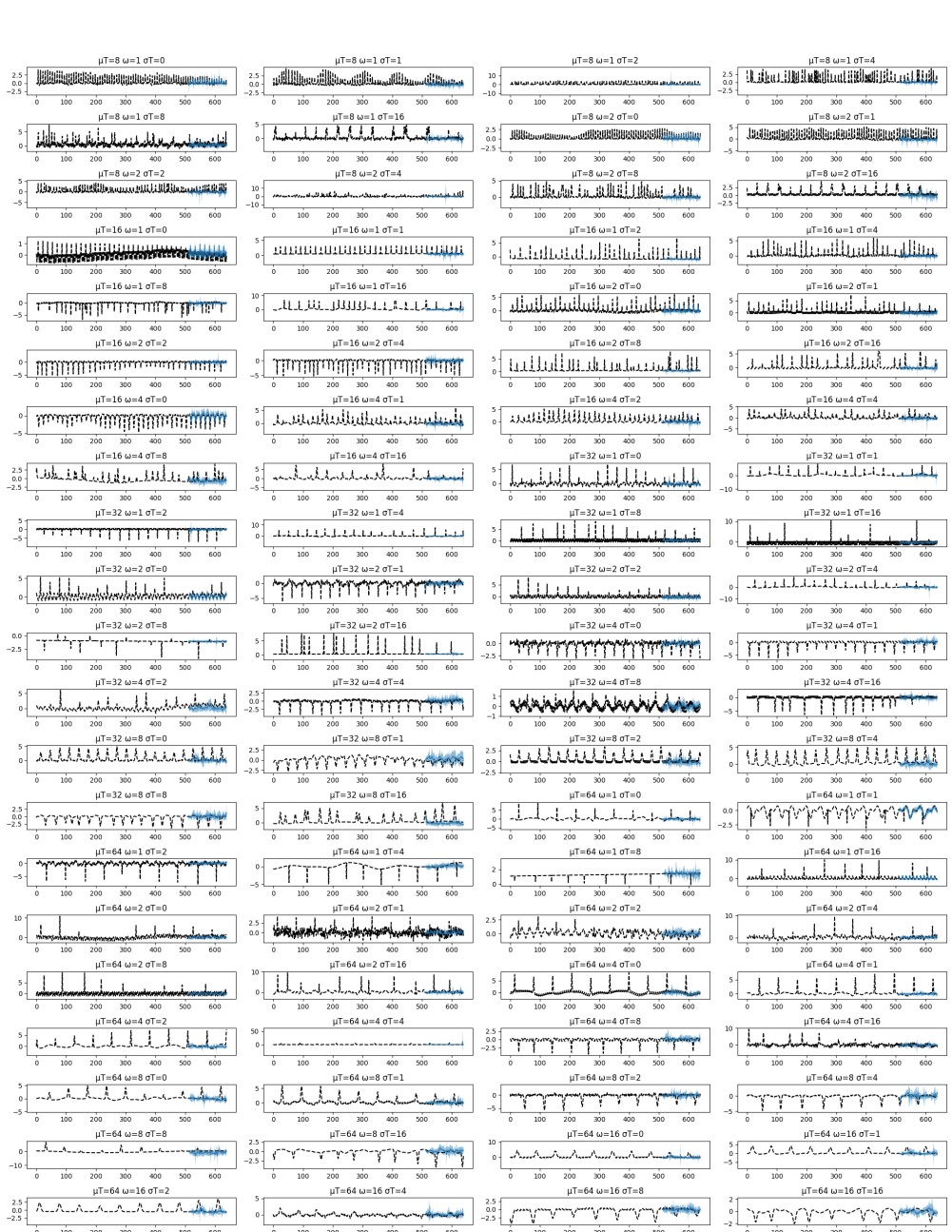

Figure 18: LAGLLAMA (200M) [Context=1024] predictions on NITH-SYNTH.

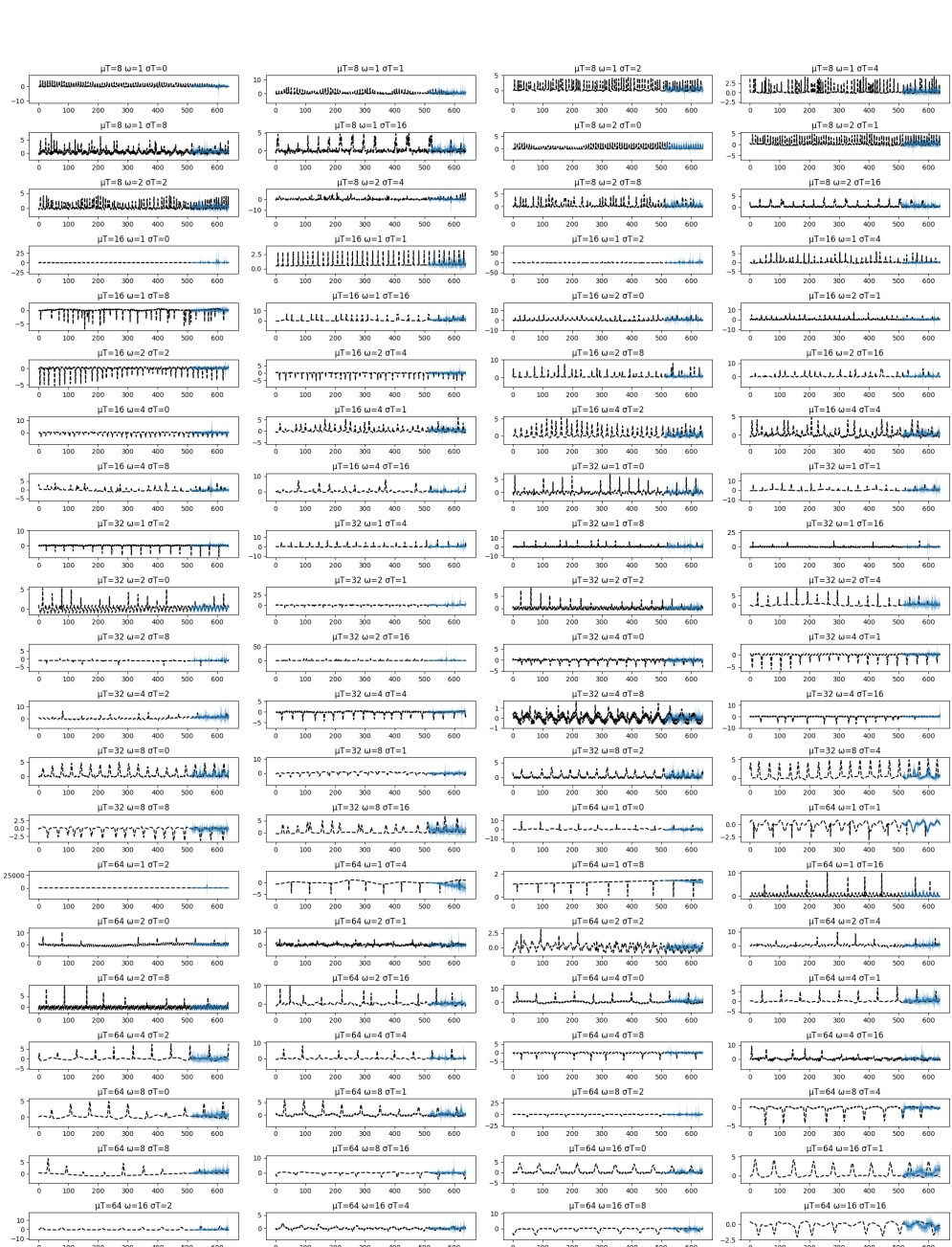

Figure 19: MOIRAI (311M) predictions on NITH-SYNTH.

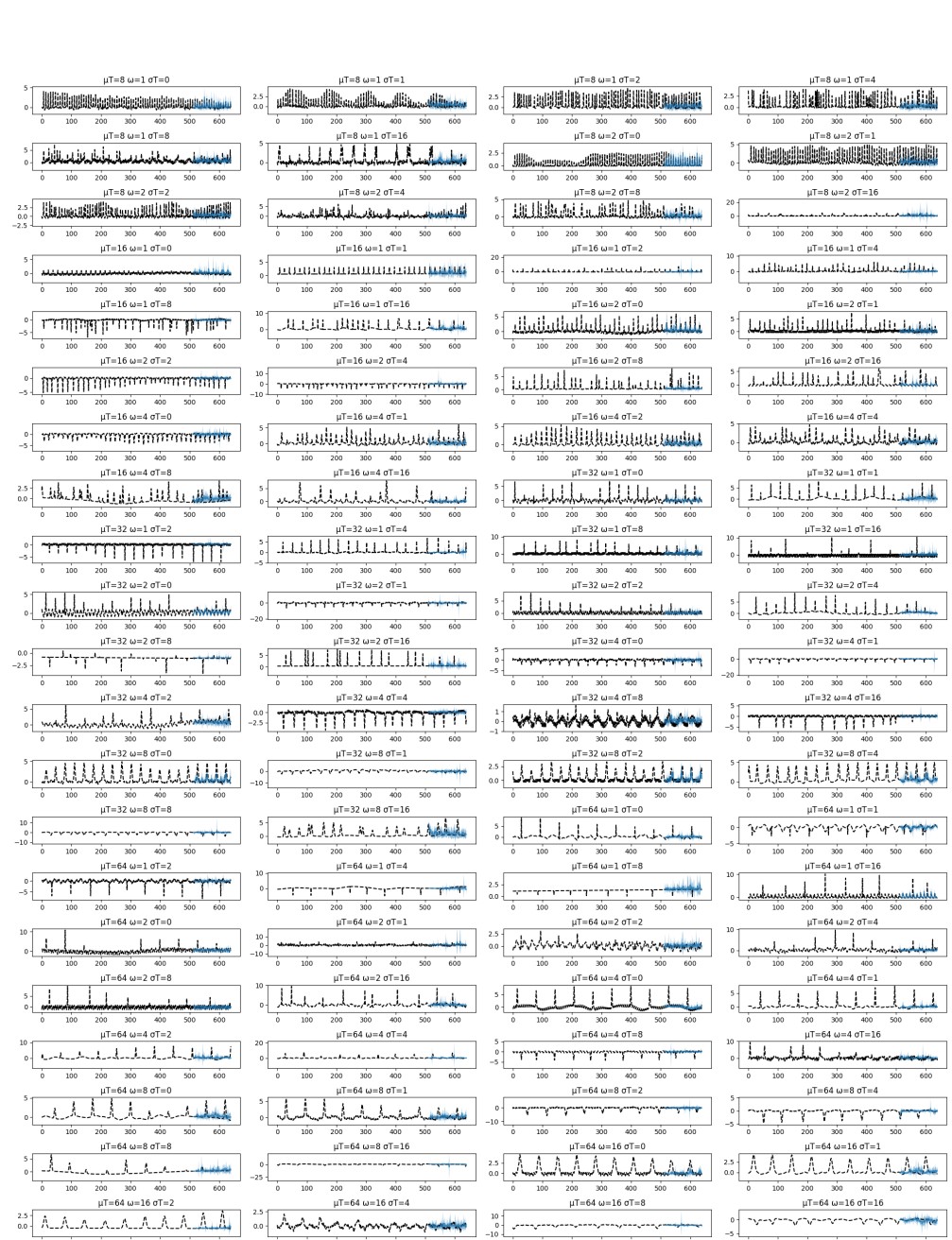

Figure 20: MOIRAI (311M) [Context=5000] predictions on NITH-SYNTH.

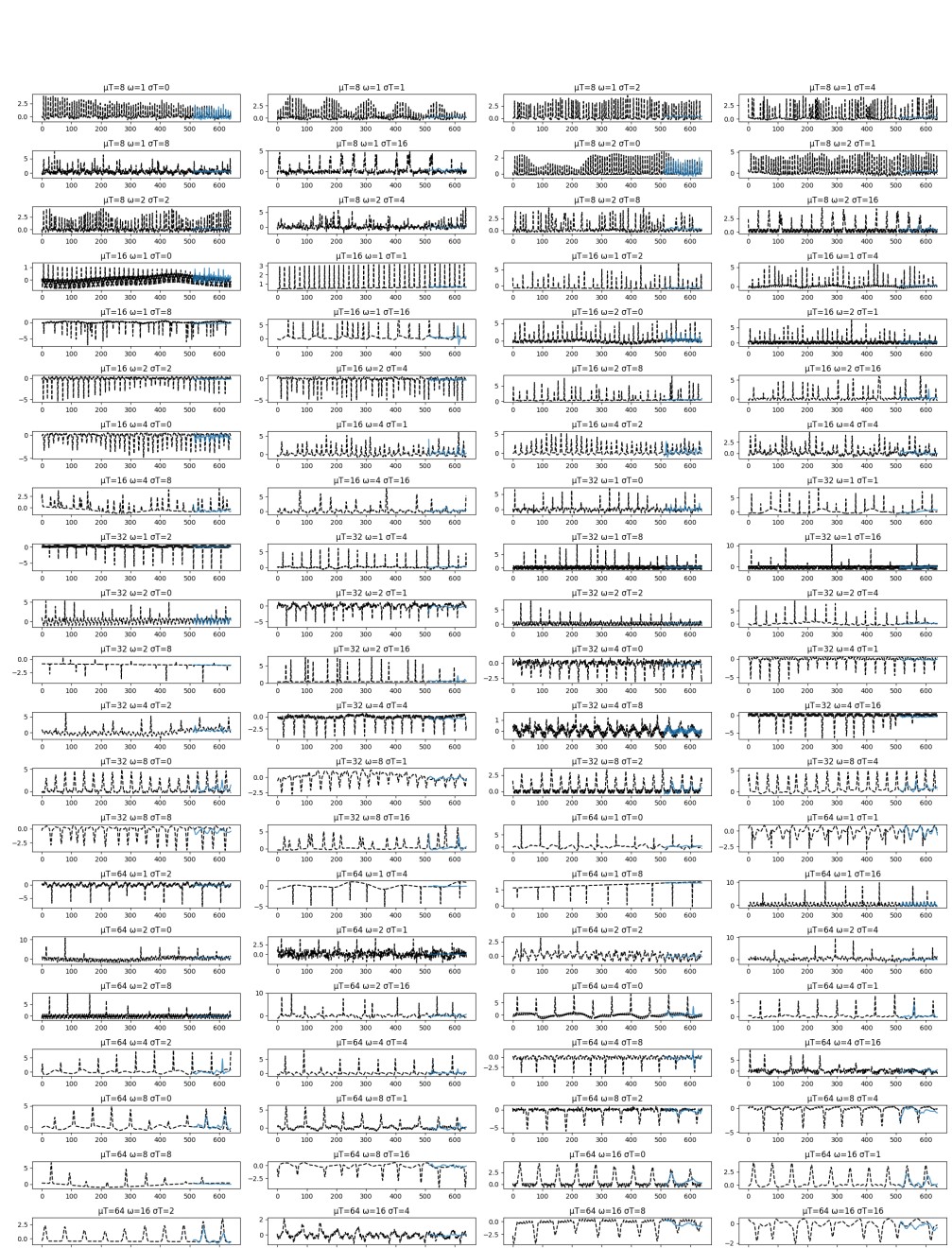

Figure 21: TIMER (67M) predictions on NITH-SYNTH.

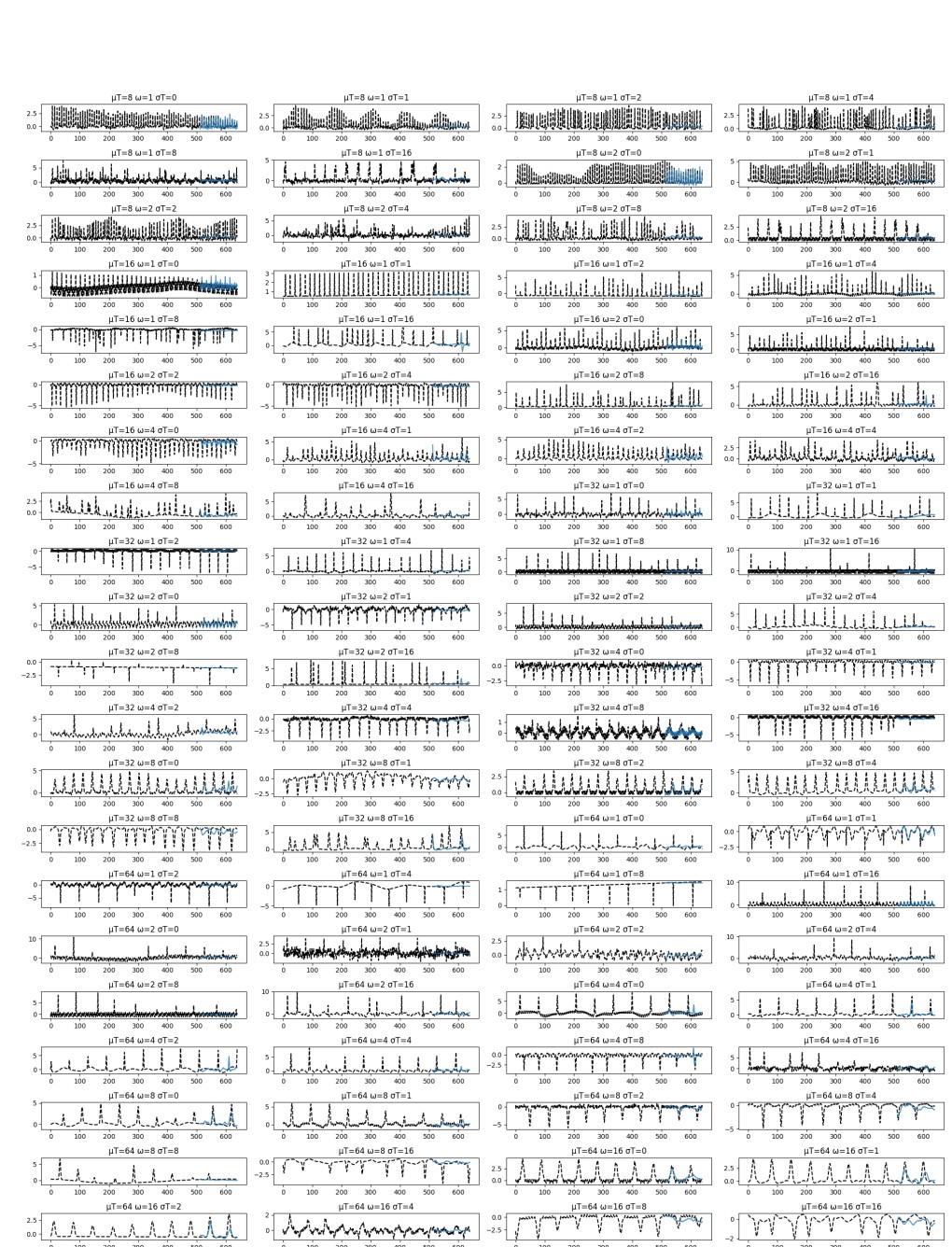

Figure 22: TIMER (67M) [Context=1440] predictions on NITH-SYNTH.

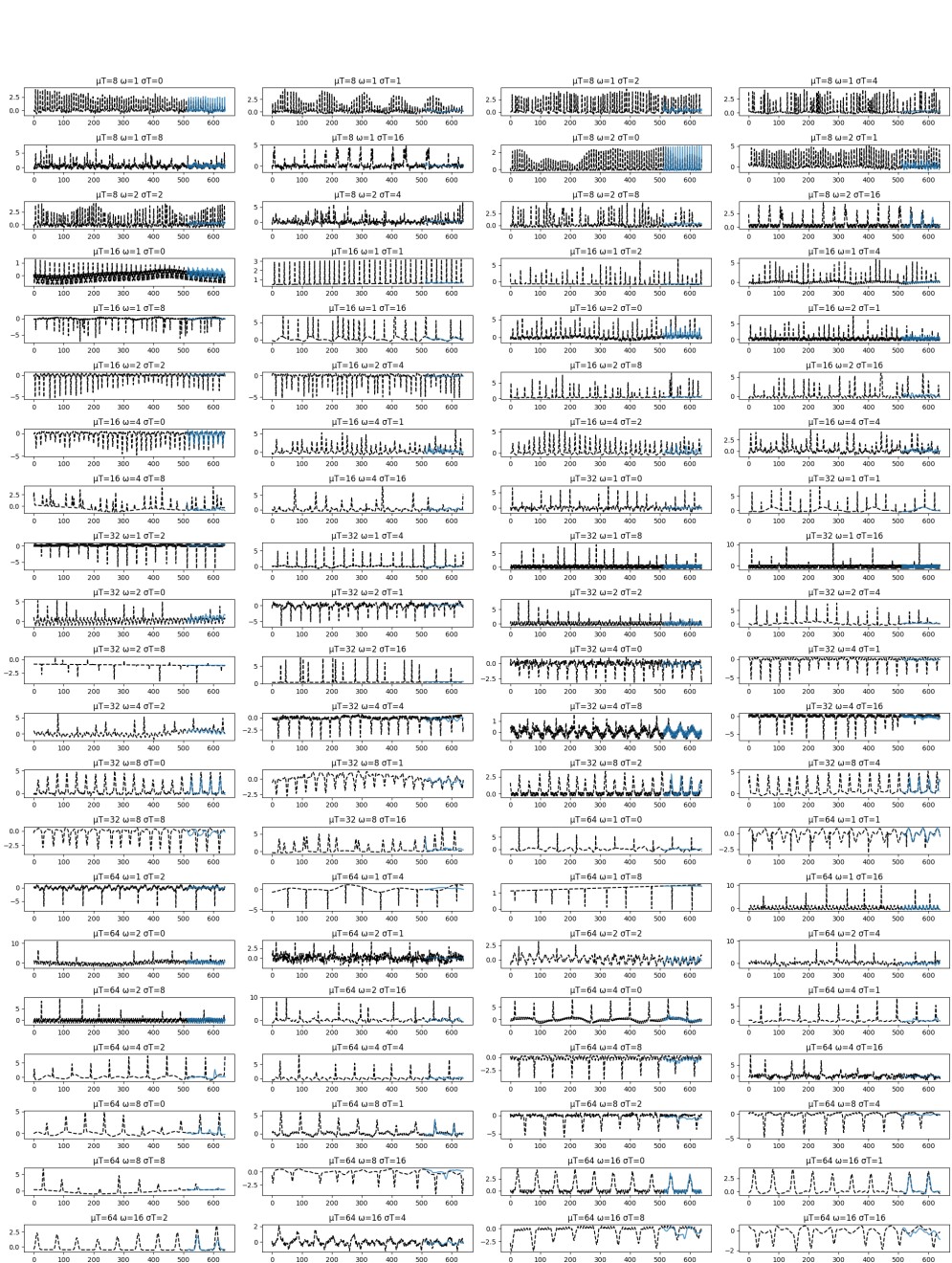

Figure 23: TimesFM (200M) predictions on Nith-Synth.

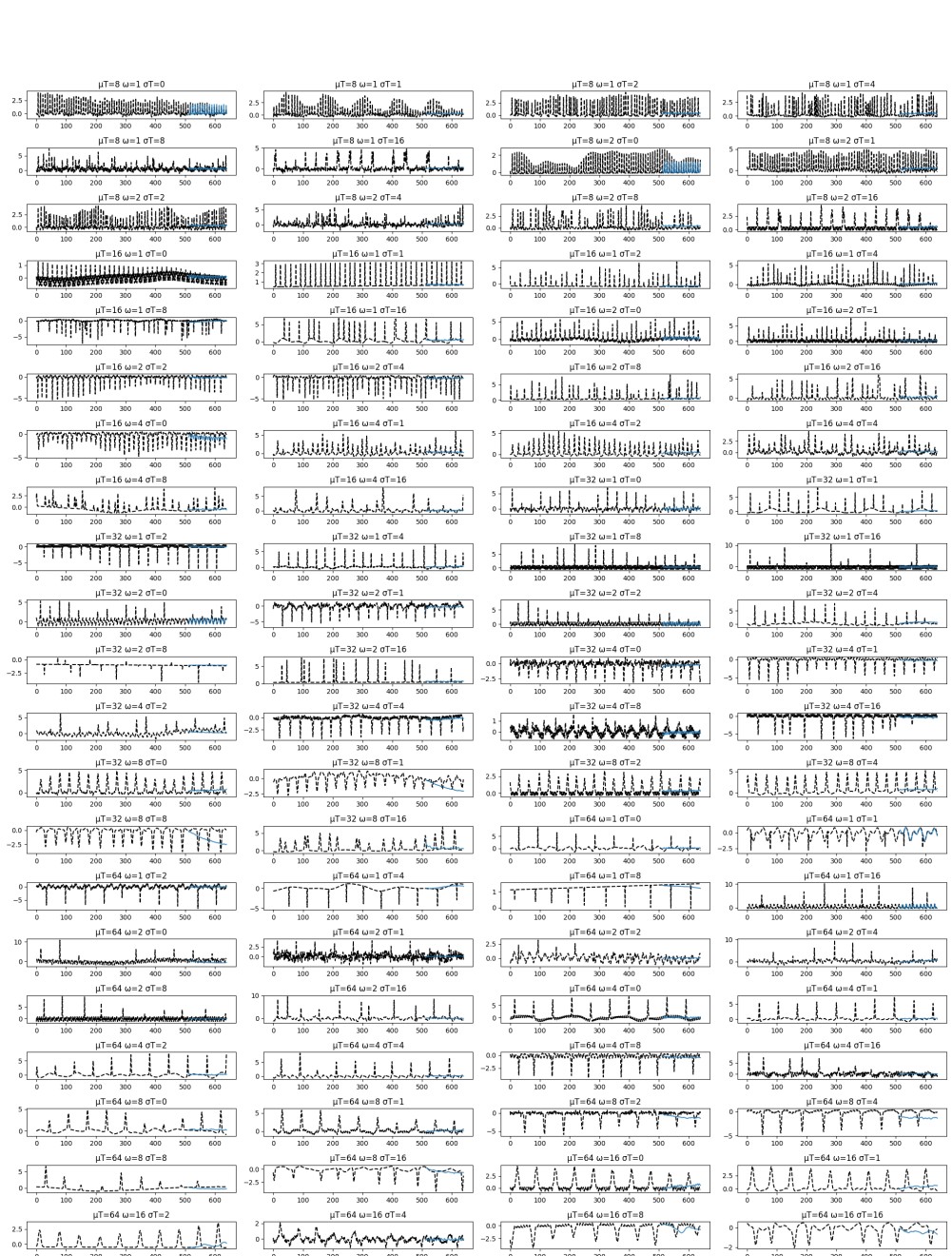

Figure 24: TINYTS (5M) predictions on NITH-SYNTH.

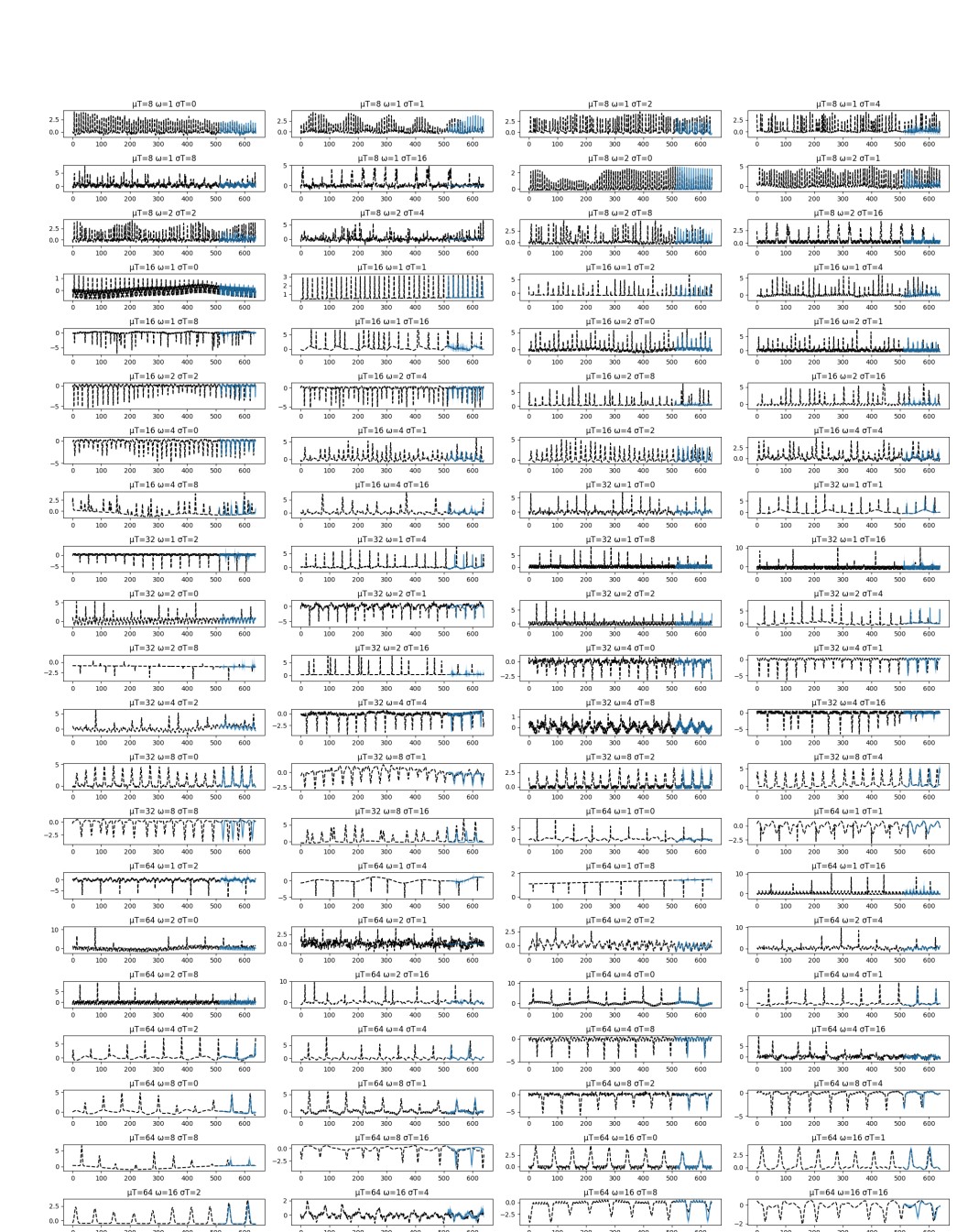

Figure 25: CHRONOS-NITH predictions on NITH-SYNTH.

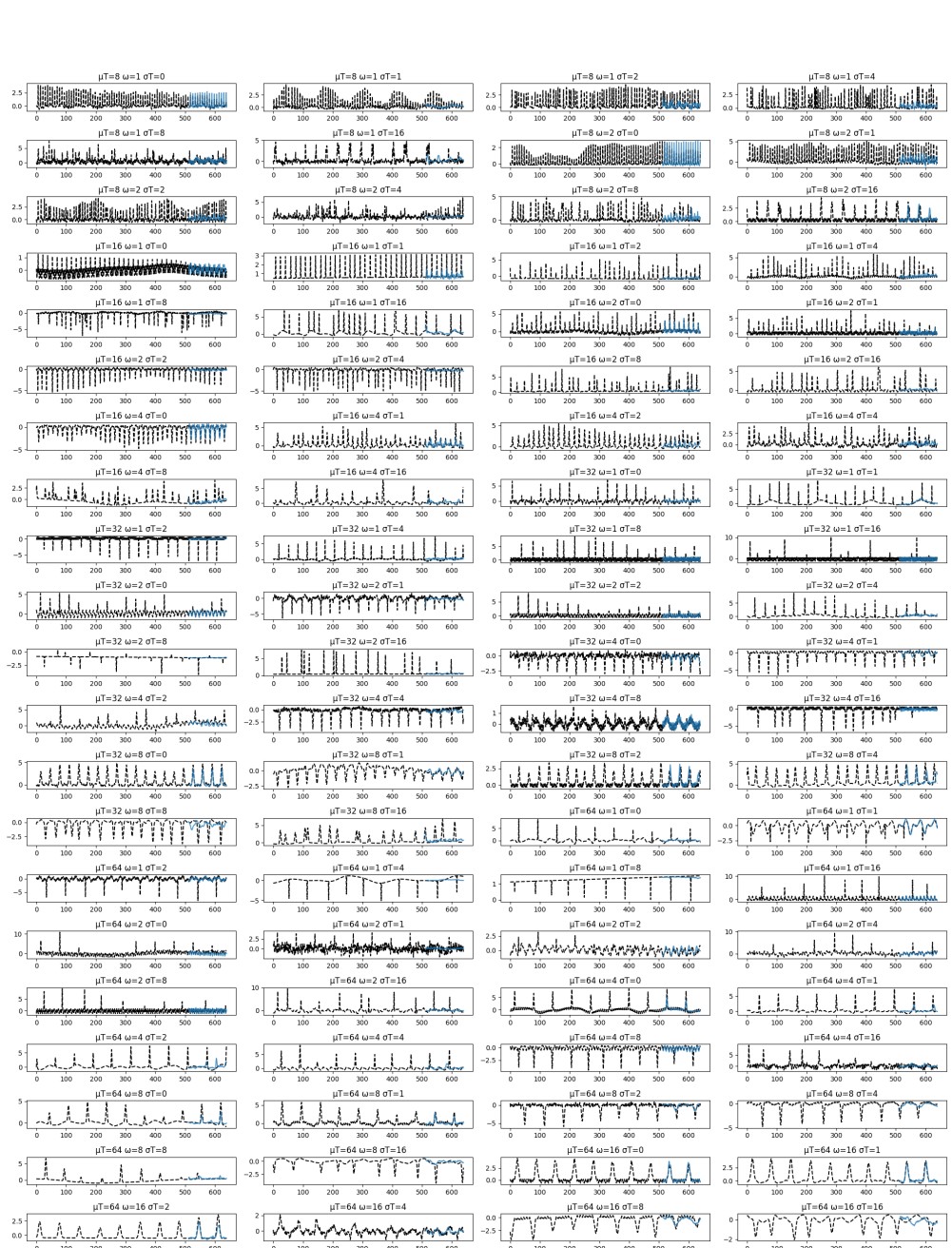

Figure 26: TIMESFM-NITH predictions on NITH-SYNTH.

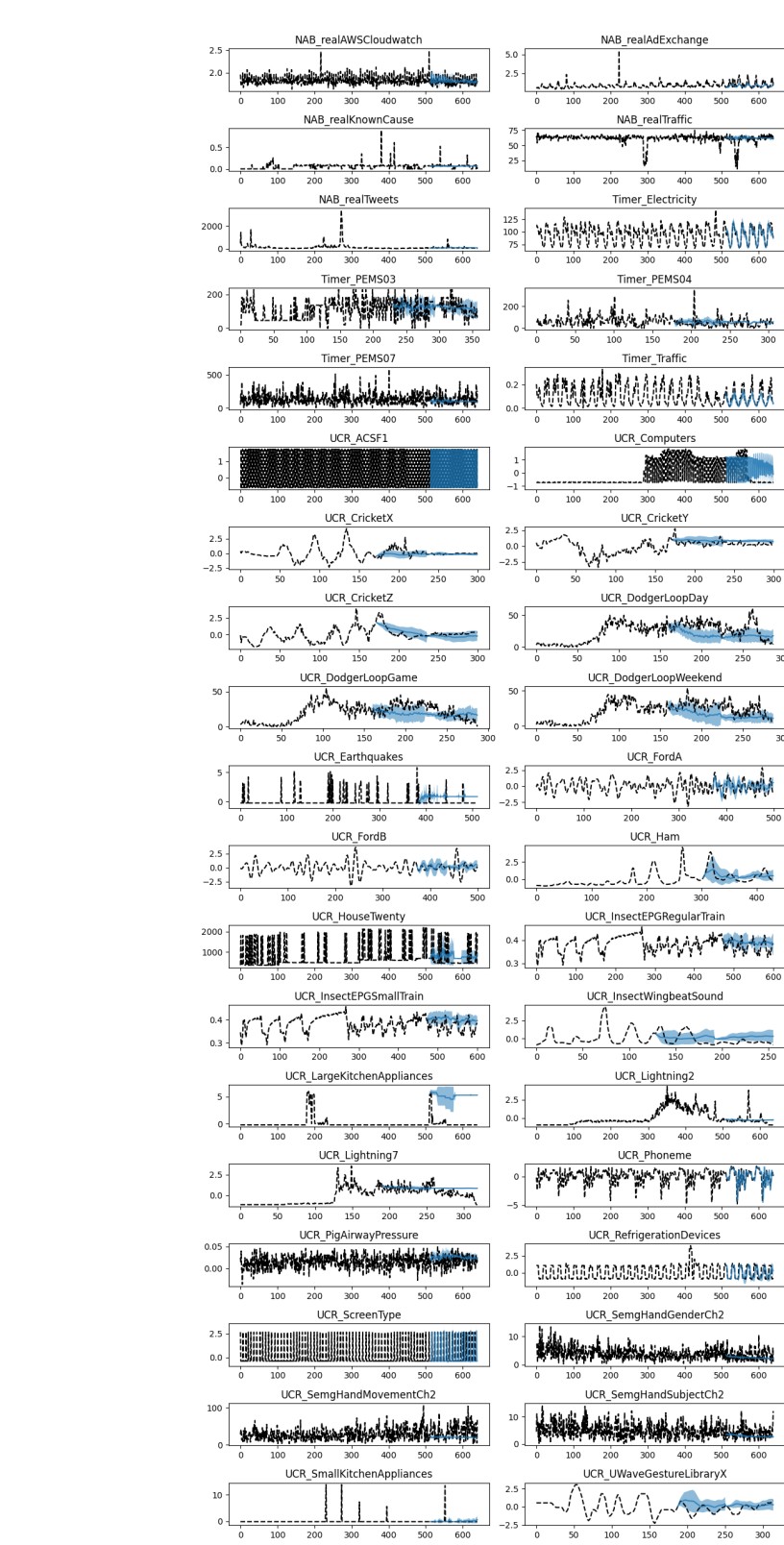

Figure 27: CHRONOS (200M) predictions on NITH-REAL.

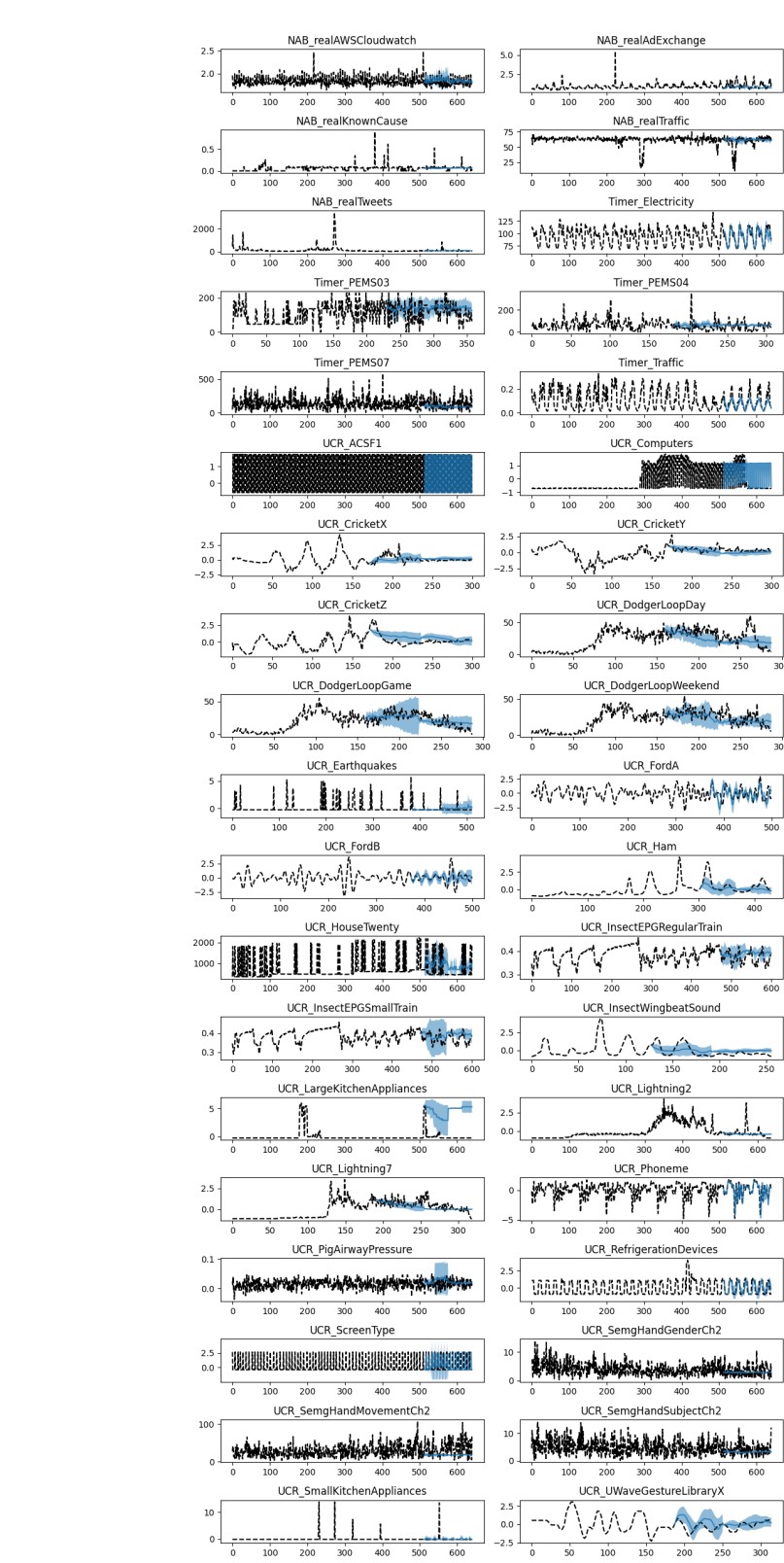

Figure 28: CHRONOS (710M) predictions on NITH-REAL.

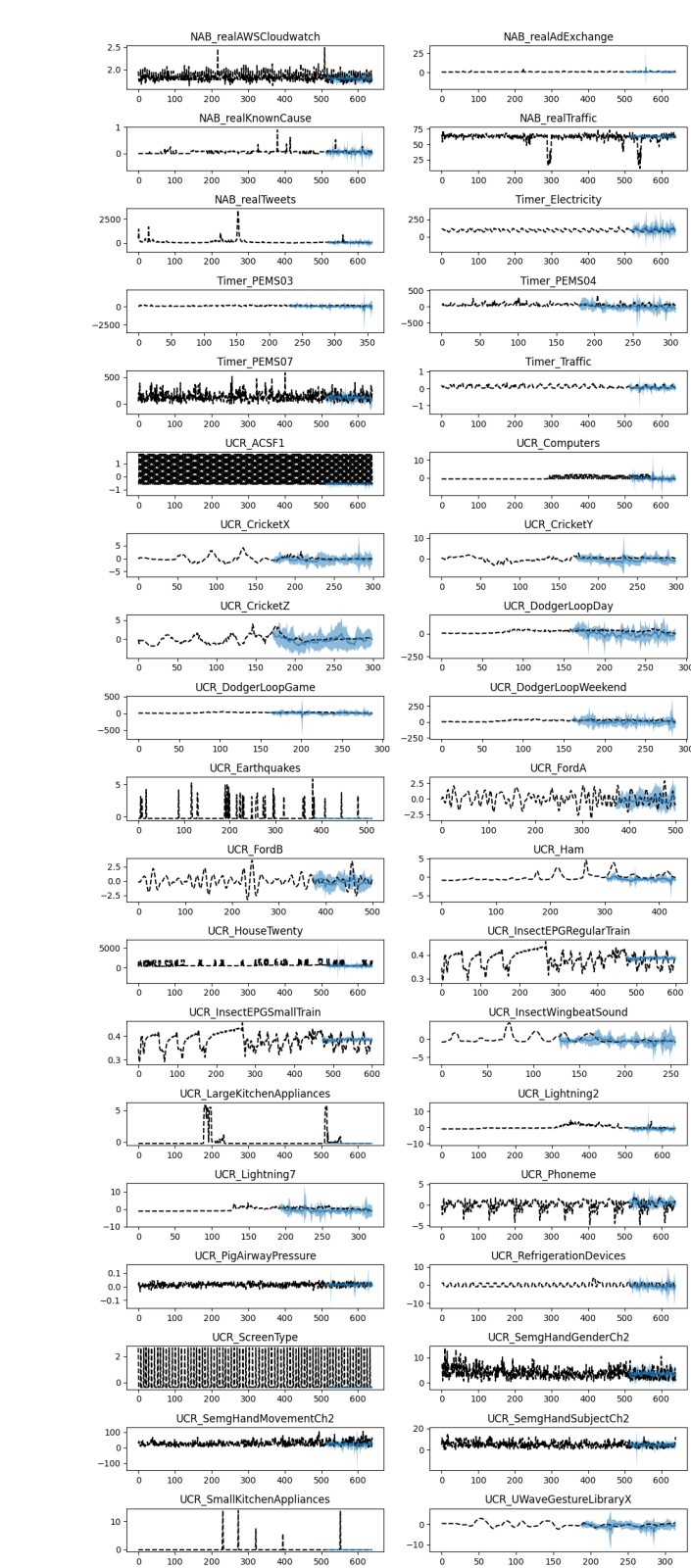

Figure 29: LAGLLAMA (200M) predictions on NITH-REAL.

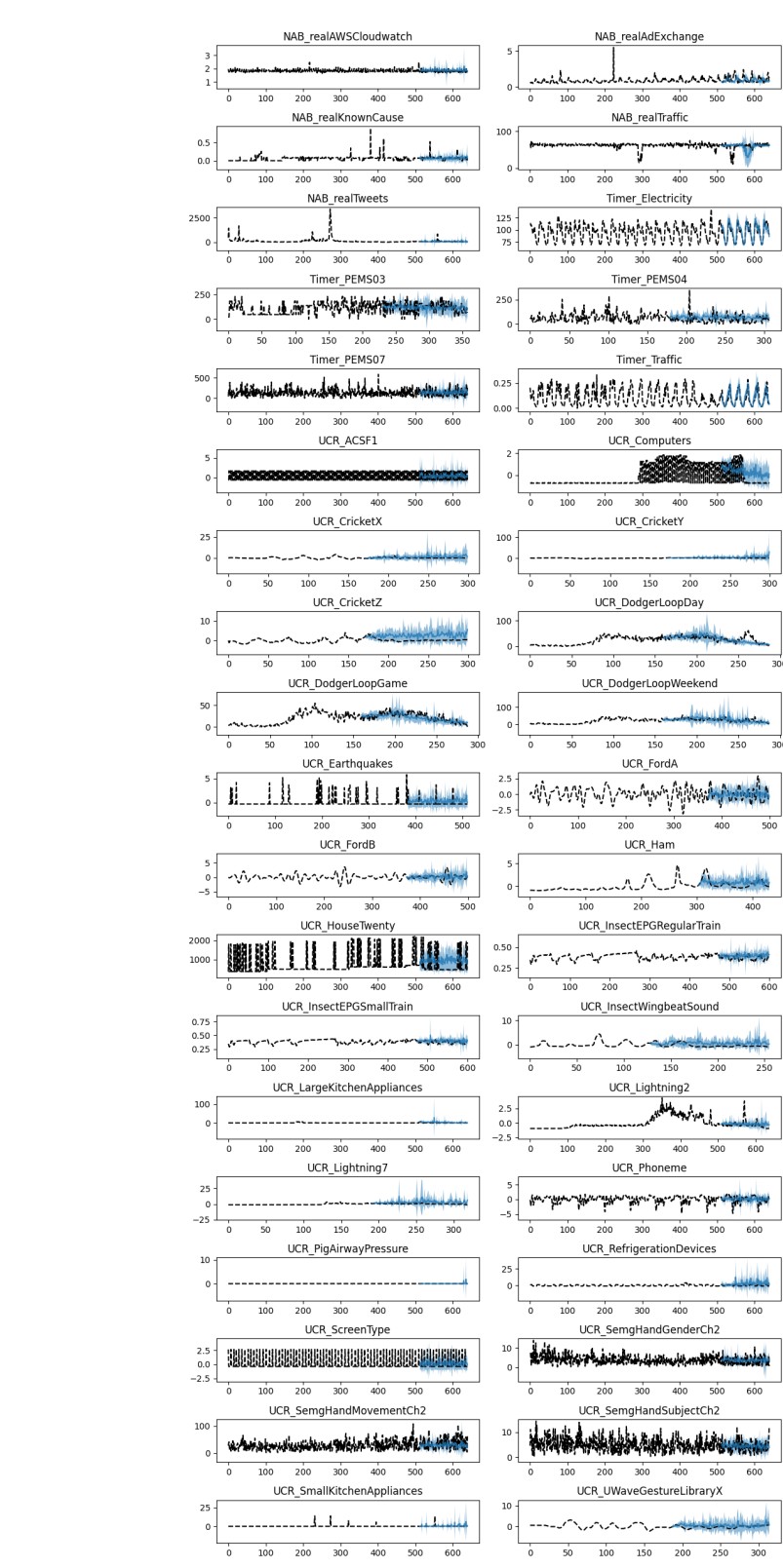

Figure 30: MOIRAI (311M) predictions on NITH-REAL.

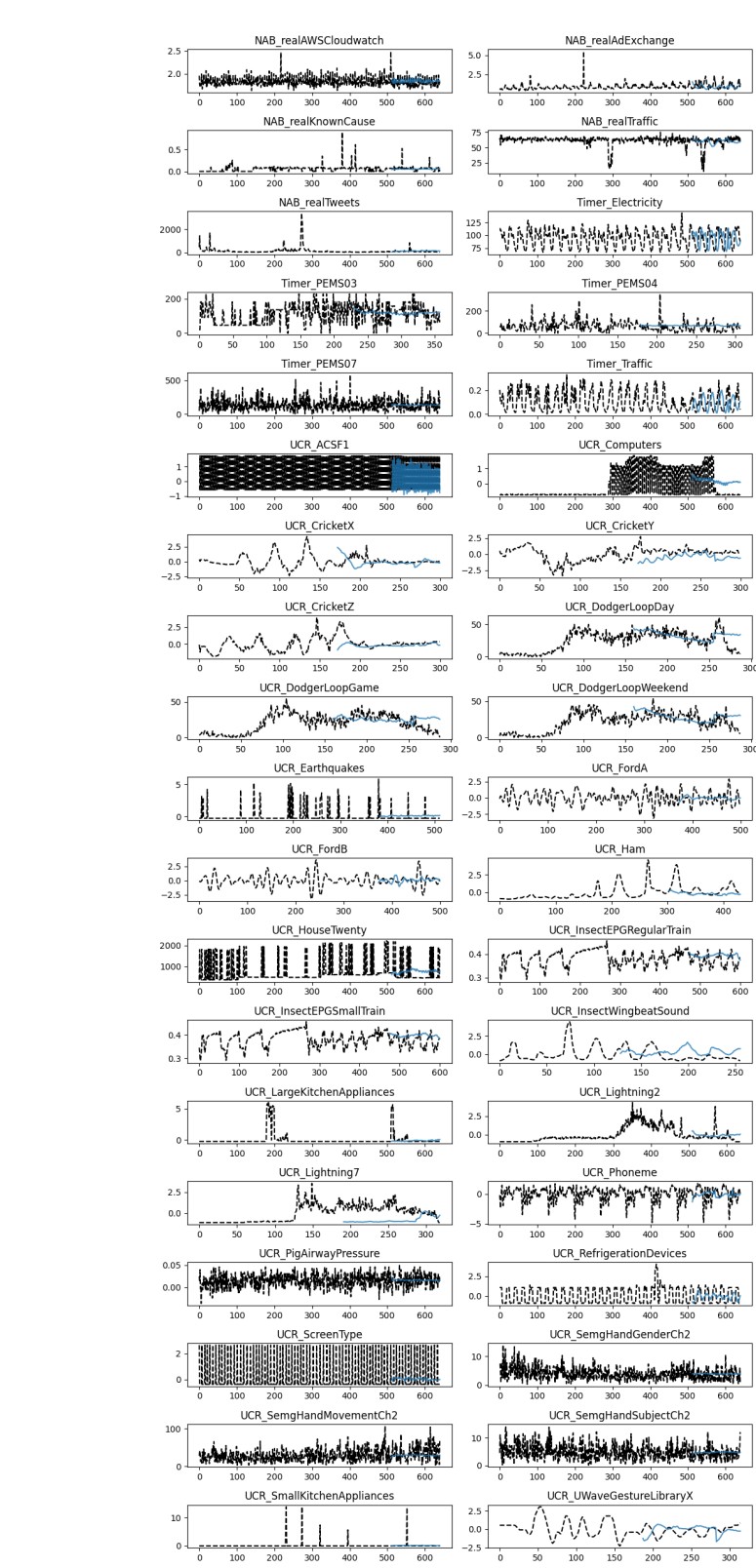

Figure 31: TIMER (67M) predictions on NITH-REAL.

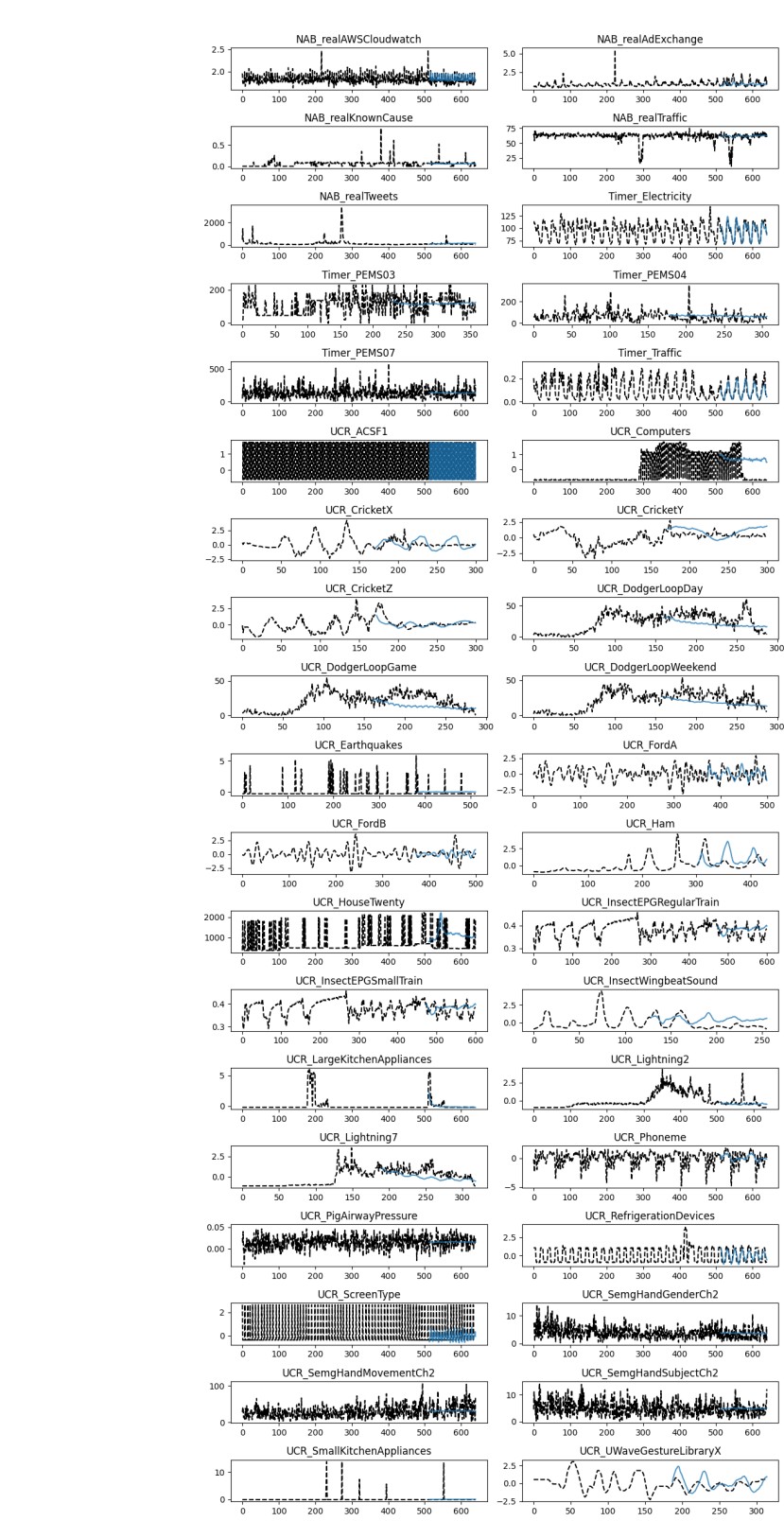

Figure 32: TIMESFM (200M) predictions on NITH-REAL.

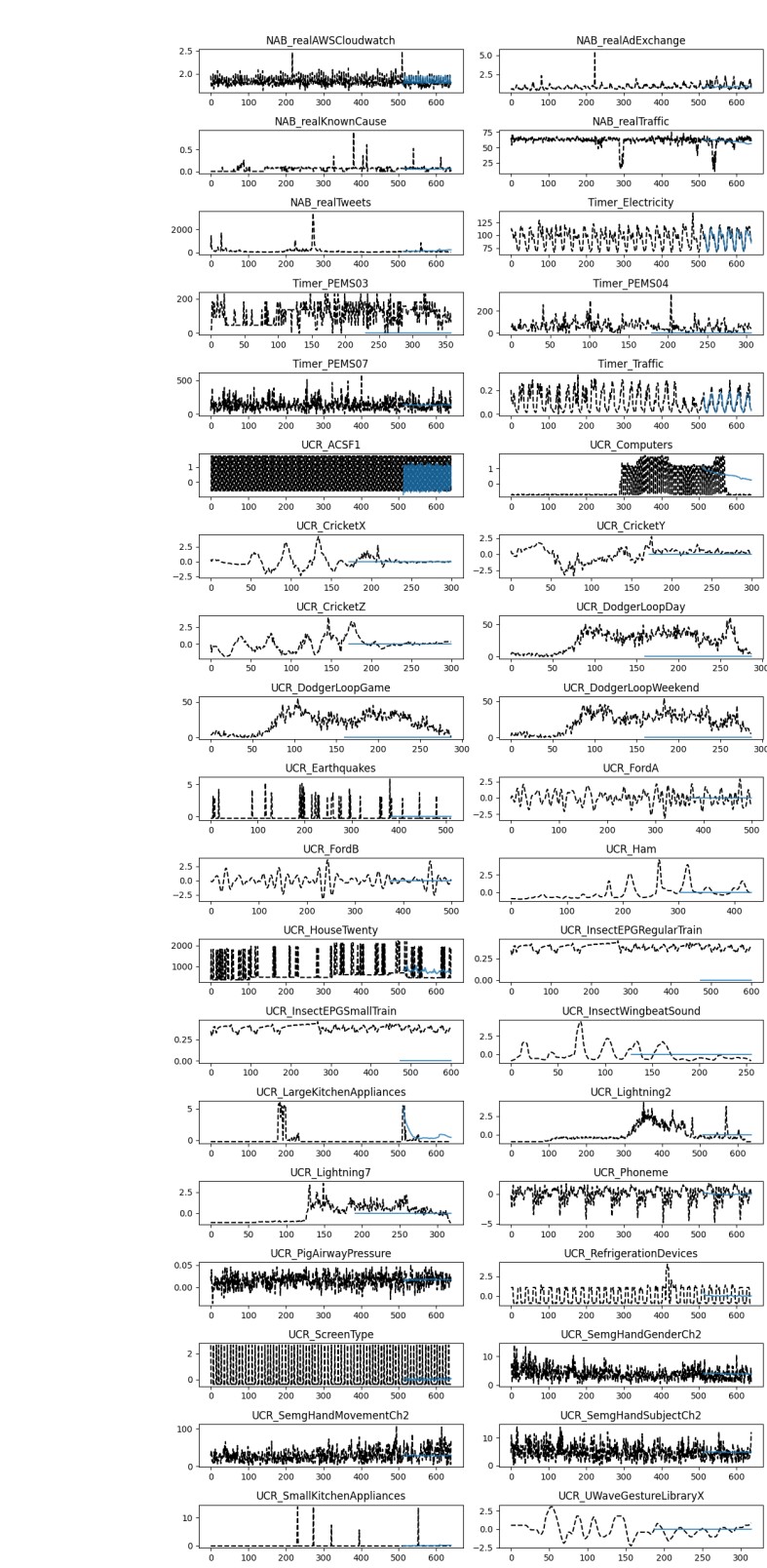

Figure 33: TINYTS (5M) predictions on NITH-REAL.

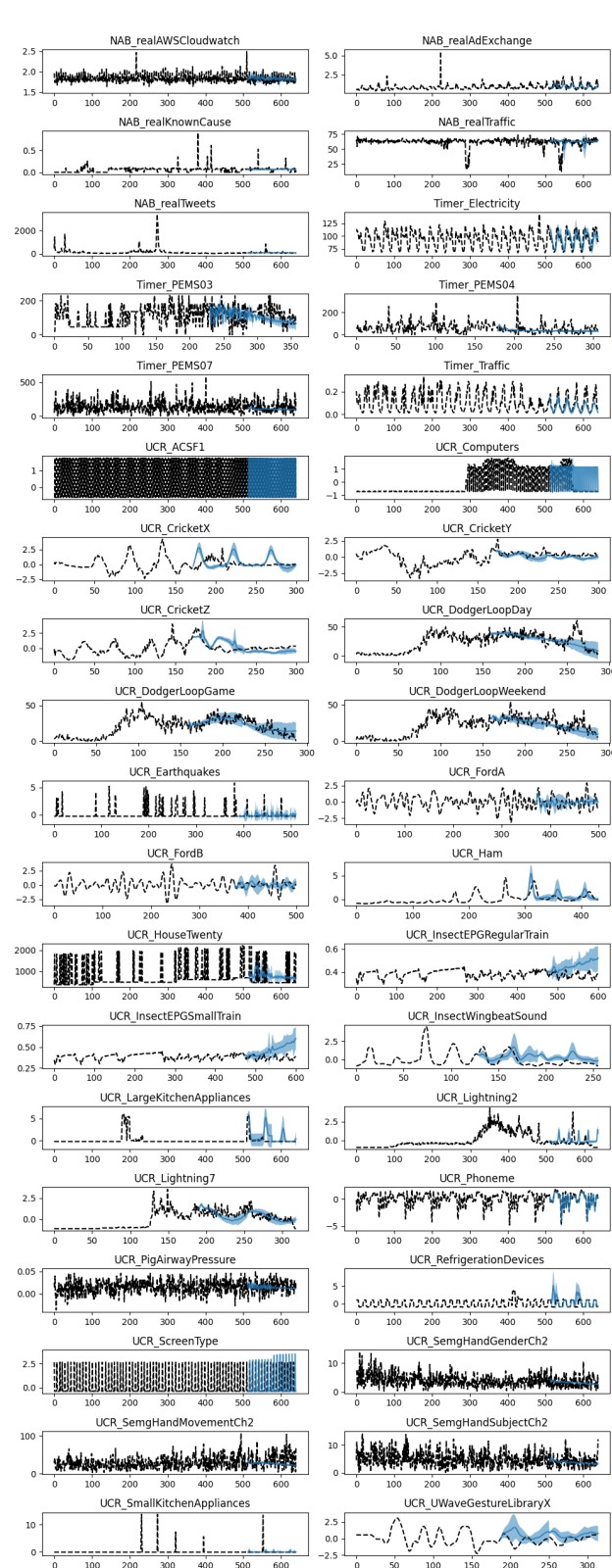

Figure 34: CHRONOS-NITH predictions on NITH-REAL.

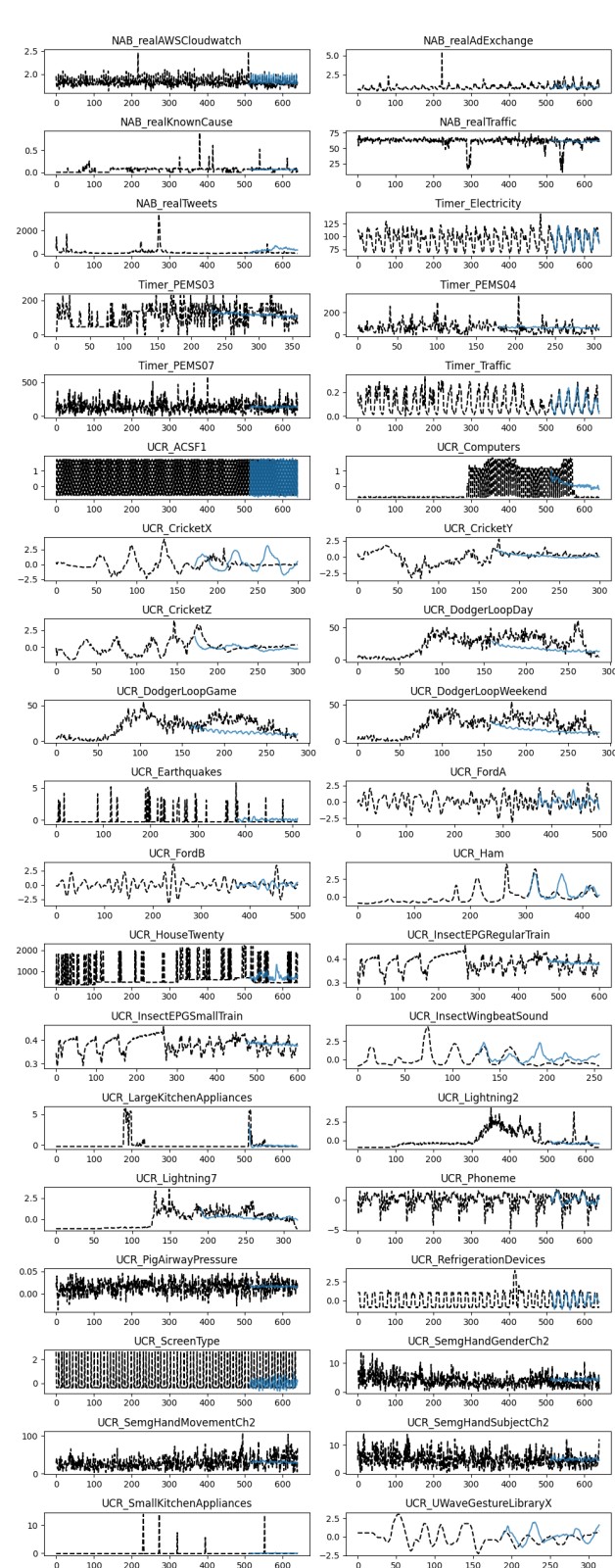

Figure 35: TIMESFM-NITH predictions on NITH-REAL.

| Datasets | Domain | $N_{test}$ | $N_{orig}$ | Length |
|---|---|---|---|---|
| NAB-realAWSCloudwatch | Machine | 6 | 17 | 4148.333 |
| NAB-realAdExchange | Machine | 6 | 6 | 1601.667 |
| NAB-realKnownCause | Machine | 4 | 7 | 4624.000 |
| NAB-realTraffic | Traffic | 5 | 7 | 2200.400 |
| NAB-realTweets | Social | 8 | 10 | 15864.250 |
| Timer-Electricity | Energy | 252 | 321 | 26304.000 |
| Timer-PEMS03 | Traffic | 1000 | 26208 | 358.000 |
| Timer-PEMS04 | Traffic | 1000 | 16992 | 307.000 |
| Timer-PEMS07 | Traffic | 1000 | 28224 | 883.000 |
| Timer-Traffic | Traffic | 614 | 862 | 17544.000 |
| UCR-ACSF1 | Energy | 87 | 100 | 1460.000 |
| UCR-Computers | Energy | 20 | 250 | 720.000 |
| UCR-CricketX | Motion | 21 | 390 | 300.000 |
| UCR-CricketY | Motion | 11 | 390 | 300.000 |
| UCR-CricketZ | Motion | 18 | 390 | 300.000 |
| UCR-DodgerLoopDay | Traffic | 5 | 80 | 288.000 |
| UCR-DodgerLoopGame | Traffic | 8 | 138 | 288.000 |
| UCR-DodgerLoopWeekend | Traffic | 8 | 138 | 288.000 |
| UCR-Earthquakes | Geophysical | 139 | 139 | 512.000 |
| UCR-FordA | Machine | 1000 | 1320 | 500.000 |
| UCR-FordB | Machine | 764 | 810 | 500.000 |
| UCR-Ham | Biomedical | 55 | 105 | 431.000 |
| UCR-HouseTwenty | Energy | 9 | 119 | 2000.000 |
| UCR-InsectEPGRegularTrain | Biomedical | 1 | 249 | 601.000 |
| UCR-InsectEPGSmallTrain | Biomedical | 1 | 249 | 601.000 |
| UCR-InsectWingbeatSound | Biomedical | 9 | 1980 | 256.000 |
| UCR-LargeKitchenAppliances | Energy | 28 | 375 | 720.000 |
| UCR-Lightning2 | Geophysical | 5 | 61 | 637.000 |
| UCR-Lightning7 | Geophysical | 1 | 73 | 319.000 |
| UCR-Phoneme | Audio | 518 | 1896 | 1024.000 |
| UCR-PigAirwayPressure | Biomedical | 8 | 208 | 2000.000 |
| UCR-RefrigerationDevices | Energy | 264 | 375 | 720.000 |
| UCR-ScreenType | Energy | 16 | 375 | 720.000 |
| UCR-SemgHandGenderCh2 | Biomedical | 361 | 600 | 1500.000 |
| UCR-SemgHandMovementCh2 | Biomedical | 270 | 450 | 1500.000 |
| UCR-SemgHandSubjectCh2 | Biomedical | 262 | 450 | 1500.000 |
| UCR-SmallKitchenAppliances | Energy | 85 | 375 | 720.000 |
| UCR-UWaveGestureLibraryX | Motion | 3 | 3582 | 315.000 |

Table 2: Statistics of different time series dataset in filtered benchmark (part 1).

| Datasets | $\mu_T$ | $\omega$ | $\sigma_T$ |
|---|---|---|---|
| NAB-realAWSCloudwatch | 21.983 | 1.868 | 50.375 |
| NAB-realAdExchange | 36.045 | 5.296 | 50.614 |
| NAB-realKnownCause | 40.820 | 2.118 | 262.003 |
| NAB-realTraffic | 22.059 | 3.270 | 30.191 |
| NAB-realTweets | 51.019 | 2.052 | 99.671 |
| Timer-Electricity | 31.126 | 11.147 | 54.411 |
| Timer-PEMS03 | 11.487 | 2.622 | 7.876 |
| Timer-PEMS04 | 11.143 | 2.488 | 7.544 |
| Timer-PEMS07 | 9.634 | 2.457 | 5.867 |
| Timer-Traffic | 30.773 | 7.760 | 37.098 |
| UCR-ACSF1 | 4.210 | 1.221 | 7.361 |
| UCR-Computers | 5.015 | 1.596 | 22.224 |
| UCR-CricketX | 36.354 | 7.219 | 23.434 |
| UCR-CricketY | 42.944 | 8.415 | 29.562 |
| UCR-CricketZ | 39.412 | 8.295 | 23.042 |
| UCR-DodgerLoopDay | 30.609 | 6.918 | 21.857 |
| UCR-DodgerLoopGame | 33.257 | 6.703 | 24.421 |
| UCR-DodgerLoopWeekend | 33.257 | 6.703 | 24.421 |
| UCR-Earthquakes | 8.090 | 1.322 | 7.626 |
| UCR-FordA | 34.920 | 10.619 | 23.311 |
| UCR-FordB | 34.605 | 10.731 | 25.759 |
| UCR-Ham | 50.239 | 8.805 | 20.840 |
| UCR-HouseTwenty | 33.228 | 3.148 | 151.309 |
| UCR-InsectEPGRegularTrain | 71.000 | 9.211 | 80.819 |
| UCR-InsectEPGSmallTrain | 71.000 | 9.211 | 80.819 |
| UCR-InsectWingbeatSound | 38.065 | 9.396 | 13.028 |
| UCR-LargeKitchenAppliances | 31.391 | 4.326 | 79.401 |
| UCR-Lightning2 | 57.524 | 3.274 | 72.528 |
| UCR-Lightning7 | 42.000 | 2.806 | 46.669 |
| UCR-Phoneme | 19.243 | 4.262 | 37.905 |
| UCR-PigAirwayPressure | 9.248 | 3.005 | 6.582 |
| UCR-RefrigerationDevices | 13.836 | 6.077 | 16.513 |
| UCR-ScreenType | 10.260 | 3.806 | 25.248 |
| UCR-SemgHandGenderCh2 | 13.916 | 2.158 | 19.442 |
| UCR-SemgHandMovementCh2 | 14.032 | 2.153 | 19.708 |
| UCR-SemgHandSubjectCh2 | 13.996 | 2.157 | 19.138 |
| UCR-SmallKitchenAppliances | 37.456 | 1.603 | 49.468 |
| UCR-UWaveGestureLibraryX | 46.364 | 12.233 | 20.486 |

Table 3: Statistics of different time series dataset in filtered benchmark (part 2).

