# OpenReview forum: "Forecasting Needles in a Time Series Haystack"
_ICLR.cc/2025/Conference — Submitted to ICLR 2025_

### Official Review · Reviewer_Cf9A · 2024-10-21

**Soundness:** 2
**Presentation:** 2
**Contribution:** 2
**Rating:** 6
**Confidence:** 3

**Summary:**

The Time Series Foundation Model (TSFM) has been a debated topic in recent years. Proper evaluation and selection of the most suitable model are also desired. The paper presents a benchmark that includes both synthetic and real data from various domains, which fills the gap in the absence of benchmarking. The paper introduces a Possion-based modeling method for generating spiky time series for synthetic data, which enables the evaluation to proceed in various conditions. The paper uses a Dynamic Time Warping based metric to define the model performance, and extensive experiments are conducted to examine the zero-shot forecasting performance of six well-established TSFMs on spiky time series.

**Strengths:**

The paper introduces a theoretical generation scheme for the spiky time series generation. The spiky time series generation follows the models of Markov chain transitioning between haystack signals that are sampled from the Gaussian process, while the needle signals are generated from a separate Gaussian process, along with outlier and rarity constraints. The paper is well-organized and has good clarity except for some minor issues in the figures.  The dataset in the benchmark is rich and full of diversity, so I believe it is useful for the evaluation of TSFMs.

**Weaknesses:**

Though the benchmark is only designed for time series forecasting, and the model performance is only evaluated in terms of the zero-shot forecasting capabilities, evaluation of other tasks might be required, e.g., classification.

Also, it would be interesting to discuss other stochastic processes for the time series generation and include an additional experiment for the ablation study.

It is hard to read in some figures in the paper, e.g., Figure 4, Figure 6-8 and Figure 14-35; I believe the fontsize should be increased.

**Questions:**

1. why the haystack distribution is set as the scaled Poisson distribution, and are there any advantages to using the scaled variant?
2. why the Gaussian process was chosen rather than other stochastic process, needs to clarify
3. what's the advantages of the method over other related generation methods [1][2], it seems like the needles signals can be instead
generated with other schemes together with the method [1].


References:
[1] https://arxiv.org/abs/2311.01388
[2 ] https://arxiv.org/pdf/2403.03698

---

> ### Author Response · Authors · 2024-11-22
> **Author Response**
>
> Thank you for your positive review! We will improve our paper following your invaluable suggestions.
>
> 1. The scaled Poisson distribution allows us to granularly control the stochasticity of the needles. Specifically, scaling decouples the Poisson distribution’s standard deviation from its mean, allowing different settings for the standard deviation.
>
> 2. Most common non-Poisson stochastic processes, (random walk, Ornstein-Uhlenbeck, and Wiener process) are instantiations of a Gaussian Process with a given kernel function. Thus, by sampling from Gaussian processes, we already include other considered stochastic processes. See the **[On Gaussian Processes]** common question for details.
>
> 3. We appreciate these additional time series generation algorithms. We chose Gaussian processes because they are more interpretable than time series generated from deep generative models. Specifically, it tests whether each time series foundation model can both capture the patterns in the Gaussian Processes’ kernel function and filter stochasticity introduced by the Poisson process.

---

> ### Comment · Reviewer_Cf9A · 2024-11-27
>
> Thanks for your clarifications! I will retain my current rating.

---

### Official Review · Reviewer_j7X4 · 2024-10-31

**Soundness:** 3
**Presentation:** 3
**Contribution:** 2
**Rating:** 3
**Confidence:** 4

**Summary:**

The authors tackle the challenge of predicting rare spikes in time series data. To help improve forecasting for these “needle” events, the authors introduce a benchmark called NITH with both synthetic and real-world datasets. They test several prominent time series foundation models and find that they generally struggle with these sharp, rare events. To improve evaluation, they propose a new metric called Scaled Dynamic Time Warping (SDTW) tailored to these spikes. Additionally, they show that pre-training models on synthetic data created to mimic real-world spiky patterns can boost model performance in this setting.

**Strengths:**

S1. The introduction of SDTW is a helpful contribution which addresses issues with existing metrics for measuring performance on the specific tasks of spiky time series forecasting.

S2. The new benchmarks (NITH-Synth, NITH-Real) capture an important subset of time-series forecasting problems, and provide a new way of evaluating and probing existing foundation models. Construction of useful benchmarks is an important way to continue to make progress on these models.

**Weaknesses:**

W1. Very strong constraints are applied to the time series setting (>8 spikes, >512 segments, etc) in order to establish the synthetic spike benchmarks. It is not clear from this work how helpful these constraints are when applying the benchmark to other real-world settings and under what settings they could be relaxed. Better establishing the importance of the ‘spike’ setting could improve this weakness.

W2. The NITH-Real dataset seems likely to have some overlap with the foundation model training sets, particularly through its use of the UCR Anomaly and NAB benchmarks. Separating out the results by benchmark and analyzing the overlap where possible could help reduce this risk.

W3. The continuous pre-training approach can lead to loss of generalization. Could the authors evaluate how the models trained on the NITH-Synth benchmark fare on other standard forecasting benchmarks? There are many existing methods for improving continual pre-training which can be applied in this setting as well to mitigate this performance difference if it exists.

**Questions:**

Q1. A significant problem with previous metrics seems to be the equal weighting of timesteps when compared to the practitioners interest in only those anomalous or spiky time-steps which follow a different frequency and magnitude. SDTW requires the ground truth needle locations as a result. Is there some way to remove this requirement, perhaps by applying a standard filter to the time-series so that MSE or DTW can be used instead?

---

> ### Author Response · Authors · 2024-11-22
> **Author Response**
>
> Thank you for your helpful review! We will update our paper to better address your concerns.
>
> W1. Thank you for bringing this up! We chose our constraints as most time series foundation models only support context windows of size 512. We chose to use more than 8 spikes as the model already cannot forecast spikes with 8 in the context, so less than 8 spikes will lead to even poorer performance. We will update our paper to better clarify how the settings were chosen.
>
> W2. Similar to Timer, we chose our datasets to prevent train-test set overlap.
>
> W3. The performance on non-spiky datasets falls. Hence, 2 separate models would be needed to capture the different evaluation-time settings.
>
> Q1. Yes. We use standard peak detection algorithms[1] as a filtering step to determine SDTW needle locations on real-world datasets without ground-truth needle locations.
>
> [1] Scikit-learn: Machine Learning in Python

---

> > ### Comment · Reviewer_j7X4 · 2024-12-02
> >
> > Thank you for responding to the review. I will retain my current rating.

---

### Official Review · Reviewer_H7Vc · 2024-11-02

**Soundness:** 2
**Presentation:** 2
**Contribution:** 2
**Rating:** 3
**Confidence:** 5

**Summary:**

The manuscript investigates the performance of learners capable of predicting spikes in time series.
Although shocks and sudden spikes are common characteristics in real-world time series (according to the authors), their main motivation is the absence of a benchmarking protocol.
To address this issue, the authors collected previously published real-world time series, created synthetic time series with those characteristics, tested some time series foundation models, and presented an extension version of DTW as a new measure.

**Strengths:**

The most important contribution is the organization of a vast amount of time series published in the literature.
This type of manuscript has great potential to be cited and is important for guiding new researchers to find more appropriate time series.
Another relevant aspect presented by the authors is the execution of foundational time series models, a hot topic nowadays, as benchmarks for their datasets.
I believe these experimental results are relevant for positioning the current research status as a foundation for future contributions. In summary, this suggests that new studies should, in some way, improve upon the presented results.
Finally, another notable strength is the volume of experiments conducted by the authors.

**Weaknesses:**

The major weakness, in my opinion, is the difficulty in clearly understanding the manuscript's goals by reading the title and abstract.
When the authors say that shocks and sudden spikes are common characteristics in real-world time series, I suppose there is substantial data available to address this problem. So, from the real-world perspective, is the authors' main contribution just the compilation of those data?
On the other hand, the authors also proposed a function to create "spiky" time series. It is an interesting proposal, but their focus is essentially on modifying the stochastic component. It is not clear whether it is also possible to modify the deterministic part of the time series, thereby varying the general behavior of the time series.
Moreover, I understand that foundational time series models are currently prominent, making it natural to use them to attract readers' attention. However, it is unclear why classical approaches were not considered in their investigation.
Another important point is the authors' choice to select MAE, MSE, and DTW as evaluation "metrics." Why were these selected? I understand they are important when comparing predicted observations, but the application presented by the authors involves not only predicting observations, but also predicting spikes. There are more appropriate "metrics" for this task that are not influenced by non-spike predictions.

**Questions:**

1 - I recommend revising all definitions and terms. For example, DTW cannot be considered a metric since it does not satisfy the triangle inequality. How does this affect the analysis? And how does it impact the proposed "metric"?

2 -The authors opted to use MAE, MSE, and DTW to evaluate their experiments. However, these may not be the most appropriate metrics. Given the focus on spikes, it’s also crucial to assess the timing of spike predictions. When comparing predictors, it is important to evaluate their ability to predict spikes accurately and the time deviation from the expected occurrence. Metrics commonly used for concept drift or change point detection might be more suitable. Why not consider more consistent evaluators?

3 -In Section 3.1.1, the authors mention that "haystack" is the base time series without spikes. Does this "base" refer to a stationary process, or could it represent any kind of time series behavior?

4 - When the authors state that "No TSFMs could accurately forecast needles across all settings," is this a result of their research? Is there a citation supporting this claim?

5 - The authors mention, "We apply our filtering algorithm." Was this algorithm proposed in this work? Where is it described, and how was it evaluated?

6 - Was "D_T" defined on Page 4?

7 - Several equations lack clarity, such as "\tau," "\hat{\tau}," and U(3,5).

8 - Figure 2 includes "pred1," which is not discussed in the text.

9 - Does Figure 3 show results for all time series?

10 - Figure 3 is not visually easy to read, especially with text closely surrounding it.

11 - The impact of adding noise to the synthetic time series should be discussed more thoroughly, particularly regarding the signal-to-noise ratio.

12 - The discussion on results often includes phrases like "We believe...," which implies some uncertainty. This phrasing gives the impression that the authors may not be entirely confident in their conclusions. Based on the results, I share the authors’ sense of uncertainty, as the conclusions do not seem fully substantiated. Consequently, the claim in the introduction about context length remains unconvincing.

13 - When creating synthetic time series, is it possible to alter the function used for generating the deterministic part of the series, independent of spike generation?

14 - In the appendices, the authors included a section on time series statistics. While it contains several plots, the statistical information does not appear substantial.

---

> ### Author Response · Authors · 2024-11-22
> **Author Response**
>
> Thank you for your constructive feedback! We will update our work to better address your concerns.
>
> 1. Thank you for this suggestion! We will update to call it similarity instead of metric.
>
> 2. You are correct, MAE MSE and DTW are not suitable metrics for evaluation. **We do not use MAE MSE and DTW for evaluation**. We empirically validate your point that MAE, MSE, and DTW are not suitable, then propose our own metric, SDTW, that combines DTW with change point detection metrics[1]. We provide additional results with simpler metrics in the **[Additional  Metrics and Baselines]** common question.
>
> 3. It could represent any time series behavior. We sample the base time series from Gaussian Processes that come from a bank of kernel functions. We agree, in some cases the best the model can do is predict the mean of past observations (ex. White noise). However, in other cases, the model can forecast underlying dynamics of the GP’s kernel function. For example, linear time series, which can be modeled, are the result of a stochastic process with the linear kernel. This work specifically studies whether the stochasticity of needles can be effectively captured by time series foundation models.
>
> 4. This is the result of our research so there is no citation.
>
> 5. Yes it is proposed in this work and described in Section 3.2.
>
> 6. D_T is set to a scaled Poisson distribution on line 249.
>
> 7-8, 10. Thank you for pointing this out. We will fix these typos and formatting.
>
> 9. Yes it shows results for all real-world time series
>
> 11. Thank you for bringing this up! We will provide further discussion on the effects of noise to model performance.
>
> 12. Thank you for these excellent suggestions! We will update our manuscript following your recommendations.
>
> 13. Yes it is possible to alter the function used for generating the deterministic part of the time series independent of spike generation.
>
> 14. Yes; we will provide updated statistics in our updated manuscript.

---

> > ### Comment · Reviewer_H7Vc · 2024-12-02
> >
> > I appreciate the authors for their efforts in addressing my comments. I will adjust my scores accordingly. While the manuscript may still require additional work to meet the standards for publication at ICLR, I believe it holds significant potential. Incorporating further experiments to better support the authors' hypothesis and refining the discussion of the conclusions could position it well for acceptance at prominent venues.

---

### Official Review · Reviewer_QJj2 · 2024-11-03

**Soundness:** 3
**Presentation:** 2
**Contribution:** 2
**Rating:** 3
**Confidence:** 4

**Summary:**

The authors study the problem of forecasting spiky time series. They proposed a dataset called Needles in the Time Series Haystack (NITH), which they used to evaluate multiple times series foundation models in zero-shot settings. They also propose a scalable way to generate spiky time series, and a novel metric to evaluate models on spiky time series. They found that most time series foundation models cannot model spiky time series. The authors also evaluate how design choices of foundation models affect their performance on spiky data.

**Strengths:**

1. The paper studies an important problem. Forecasting or predicting spiky data is indeed understudied.
2. The authors conduct a **lot** of experiments-- with multiple times series foundation models, and their design choices
3. The author propose a scalable way to generate spiky time series, and evaluate models on forecasting spiky data

**Weaknesses:**

1. **Clarity**: The paper introduces a lot of "novel" things, and mathematical instruments to model and evaluate time series models on spiky data. There's a lot of notation which is hard to follow ($\alpha, \beta, \theta, \omega$, to name a few) as sometimes symbols apply prior to their usage. For example, it is particularly hard to follow 219 -- 225, where it's unclear what $\alpha$ and $\beta$ are until it is introduced in Algorithm 1.
2. **Too many "novel" things, but their :** The authors propose too many new things, that it is hard to keep track. Moreover, it seems that the authors miss a lot of basic things and instead over-complicate their proposed solutions. For example, [1] proposes a simple way to inject spikes in time series which was not considered. For the "haystack", the authors use Gaussian Processes, when simpler time series can be generated using polynomial or linear trends, periodicity (sine waves), and some noise (Gaussian or Random walks). Similarly, there has been a lot of work in time series anomaly detection on better metrics to evaluate time series of similar natures, for example [2]. Moreover, simpler metrics such as $\alpha^{T}(Y - \hat{Y})$, where $\alpha$ is a vector which gives more weight to needles rather than the haystack.
3. **Baselines and related work:** The authors only consider time series foundation models. They do not consider any statistical models such as ARIMA, or models which are designed to predict spiky time series. Moreover, the authors do not consider any prior work on modeling or evaluating spiky time series, e.g. [3, 4, 5, 6]. Also, the authors continually pre-train pretty large models on NITH, perhaps smaller models can be fine-tuned more effectively on spiky time series.
4. **Normalization:** All time series models including foundation models use some form of normalization (e.g. Chronos uses mean scaling while TimesFM uses RevIN), and I believe that this influences the forecasting performance. This is perhaps the most component of the design space of these models.
5. **Lags and lack of connection to a real-world application:** The authors mention (random) lag multiple times in the paper, but it is unclear to me why is this an important consideration for spiky time series? I feel this is also where connection to a real-world problem (e.g. energy load prediction) would be helpful. For example, in these cases it may be more important for to predict whether there will be spikes, or the number of spikes and the amplitude of the spike, each of which is a different problem (classification or regression), and can be evaluated with different metrics.
6. **Loss function:** Most TSFMs are trained to predict some notion of conditional mean (MSE) or median (MAE) of a time series. These loss functions are not appropriate for training models that can be expected to do well on spiky data, thus the fact that TSFMs don't perform too well is not surprising. On the other hand, I feel that the authors could have spent more time on this important aspect.
7. **Memorization:** The authors claim that the benchmark tests a model's ability to memorize spikes. Could the authors provide more information and evidence on why this is the case?

## Minor
1. In line 380, I am not sure what "struffle" means
2. I think the results in Figure 7 may not be comparable because the encoder-only, decoder-only and encoder-decoders models have different number of parameters, so it is unclear if the differences are due to the difference in # of parameters or due to the inductive biases of these models.
3. It is not immediately clear to me what is a good value of the scaled DTW.
4. In line 316, I suspect that the authors have mistakenly annotated the wrong predicted time series (they have interchanged Pred 2 and 3).


### References
1. Goswami, M., Challu, C. I., Callot, L., Minorics, L., & Kan, A. Unsupervised Model Selection for Time Series Anomaly Detection. In The Eleventh International Conference on Learning Representations.
2. Paparrizos, J., Boniol, P., Palpanas, T., Tsay, R. S., Elmore, A., & Franklin, M. J. (2022). Volume under the surface: a new accuracy evaluation measure for time-series anomaly detection. Proceedings of the VLDB Endowment, 15(11), 2774-2787.
3. Fadlallah, B., Chen, B., Keil, A., & Príncipe, J. (2013). Weighted-permutation entropy: A complexity measure for time series incorporating amplitude information. Physical Review E—Statistical, Nonlinear, and Soft Matter Physics, 87(2), 022911.
4. López, Manuel Zamudio, Hamidreza Zareipour, and Mike Quashie. "Forecasting the Occurrence of Electricity Price Spikes: A Statistical-Economic Investigation Study." Forecasting 6.1 (2024): 115.
5. Fatone, L., Mariani, F., Recchioni, M. C., & Zirilli, F. (2012). The analysis of real data using a stochastic dynamical system able to model spiky prices.
6. Zhang, X., Jin, X., Gopalswamy, K., Gupta, G., Park, Y., Shi, X., ... & Wang, Y. (2022). First de-trend then attend: Rethinking attention for time-series forecasting. arXiv preprint arXiv:2212.08151.

**Questions:**

1. Is DTW or it's scaled version as you have proposed differentiable? If not, how are models trained using these as a loss function?
2. What loss function is used to continually pre-train Chronos and TimesFM?

---

> ### Author Response · Authors · 2024-11-22
> **Author Response**
>
> Thank you for your insightful review! We will improve our benchmark following your recommendations.
>
>
> 1. We will create a table with mathematical notations to make this more clear. $alpha$ and $beta$ are scaling parameters which scale the needle signal to be much larger than (>>) the haystack signal. $omega$ is the width of the needle. $theta$ is the flipping parameter.
>
> 2. We address this in the **[On Gaussian Processes]** common question. We will update our manuscript to better organize our contributions: 1) we propose a real and synthetic benchmark of spiky time series, 2) we provide a synthetic dataset generation algorithm, 3) we provide a new evaluation metric measuring spiky time series prediction, and 4) we benchmark different foundation models and techniques on spiky time series data. We provide your requested metrics in the **[Additional Metrics and Baselines]** common question.
>
> 3. We provide your requested ARIMA baseline in the **[Additional Metrics and Baselines]** common question.
>
> 4. We recommend median scaling for this problem.
>
> 5. Our application is exactly what you listed - predicting the number and amplitude of spikes. To this end, our evaluation metric is a regression metric between predicted and true curves. We provide your requested anomaly detection metrics in the **[Additional Metrics and Baselines]** common question.
>
> 6. You are correct that MSE encounters significant challenges when predicting Poisson processes. We identify alternative loss functions proposed for this also do not work well. We will expand discussion on this section.
> 7. By memorization, we mean the model should memorize the expected durations between spikes. Sorry for the confusing wording! We will update our work to clarify this.
>
> Minor: Thank you for these comments. We will update our paper to incorporate these helpful suggestions!
>
> Q1: DTW is not differentiable as it is discontinuous when selecting the optimal path; however, we experimented with a soft relaxation: DILATE. The models were trained on either MSE or Cross Entropy loss for patch or sample tokenization respectively.
>
> Q2: We use Cross Entropy with Chronos and MSE with TimesFM.

---

> > ### Comment · Reviewer_QJj2 · 2024-11-23
> > **Thanks for the rebuttal!**
> >
> > Dear Authors,
> >
> > Thank you for addressing my comments and for the revisions you have made (or plan to make) to the paper. After reviewing your responses, I must admit that I still lean toward recommending rejection at this stage.
> >
> > In my view, the article requires further refinement before it is ready for publication. For instance, certain design choices, such as the synthetic data generation process, could be simplified. Additionally, incorporating multiple robust baselines (beyond ARIMA) would enhance the rigor of the experimental evaluation.
> >
> > That said, I appreciate the progress made during the rebuttal and encourage the authors to take the feedback into account to continue improving the manuscript.
> >
> > Thank you!

---

### Author Response · Authors · 2024-11-22
**Common Response**

We thank all the reviewers for their invaluable feedback, which we will use to improve our paper.

**[On Gaussian Processes]**

Our choice in Gaussian processes as the base needle and haystack signals encompasses existing stochastic processes such as random-walks, Ornstein-Uhlenbeck, or Wiener processes. Specifically, we chose the GPs from a kernel bank of popular Gaussian processes following established existing works[2].

**[Additional Metrics and Baselines]**

We implement metrics from the anomaly detection field: R-F1 and R-VUC[1]. We use the same settings for R-VUC as the original paper. We add the additional ARIMA baseline to test the effectiveness of statistical methods with the frequency set to each dataset’s corresponding frequency. We compute R-VUC and SDTW by using peak detection algorithms to convert the predicted signal into anomaly scores. RF-1 gives uninformative results, because the needle locations do not overlap. Both R-VUC and SDTW agree that TimesFM is the best model.

```
              | VUC      | RF-1     | SDTW
--------------+----------+----------+-------
TimesFM       | 3.58E-04 | 8.33E-02 | 1.98
Moirai        | 1.66E-03 | 5.52E-02 | 2.26
Timer         | 1.87E-03 | 5.55E-02 | 2.27
Chronos(base) | 7.82E-04 | 8.00E-02 | 2.35
Chronos(small)| 1.82E-03 | 8.17E-02 | 2.38
Lagllama      | 2.08E-03 | 5.94E-02 | 2.49
Arima         | 3.86E-03 | 2.84E-02 | 2.50
```

[1] Volume under the surface: a new accuracy evaluation measure for time-series anomaly detection.
[2] Chronos: Learning the Language of Time Series

---

### Meta-Review · Area_Chair_dZPv · 2024-12-19

**Metareview:**

This paper has been evaluated by 4 knowledgeable reviewers. 3 of them recommended rejecting it (all 3 are straight rejection scores), while one gave it a marginal accept. The limitations pointed out in the reviews are of a major nature and cannot be addressed quickly. The prevalent opinion of the reviewers is that the paper is not ready for publication in its current form, even after the improvements implemented as part of the rebuttal, however the reviewers agreed that the work has potential.

**Additional Comments On Reviewer Discussion:**

In the discussion with the authors, reviewers pointed out that certain design choices, such as the synthetic data generation process, could be simplified. Additionally, incorporating multiple additional baselines (beyond ARIMA) would enhance the rigor of the experimental evaluation. The tone of the reviews has indicated that even though the manuscript requires additional work to meet the standards for publication at ICLR, it holds good potential. So expanding the scope of experiments to better support the posed hypotheses and refining the discussion, could position it for acceptance at prominent future venues.

---

### Decision · Program_Chairs · 2025-01-22

Reject